# Adipose Tissue-Derived Extracellular Vesicles Contribute to Phenotypic Plasticity of Prostate Cancer Cells

**DOI:** 10.3390/ijms24021229

**Published:** 2023-01-08

**Authors:** Allison Mathiesen, Bronson Haynes, Ryan Huyck, Michael Brown, Anca Dobrian

**Affiliations:** Department of Physiological Sciences, Eastern Virginia Medical School, Norfolk, VA 23501, USA

**Keywords:** extracellular vesicles, endothelial cells, obesity, *TWIST1*, PC3ML, miRNA, epithelial-to-mesenchymal transition, inflammation

## Abstract

Metastatic prostate cancer is one of the leading causes of male cancer deaths in the western world. Obesity significantly increases the risk of metastatic disease and is associated with a higher mortality rate. Systemic chronic inflammation can result from a variety of conditions, including obesity, where adipose tissue inflammation is a major contributor. Adipose tissue endothelial cells (EC) exposed to inflammation become dysfunctional and produce a secretome, including extracellular vesicles (EV), that can impact function of cells in distant tissues, including malignant cells. The aim of this study was to explore the potential role of EVs produced by obese adipose tissue and the ECs exposed to pro-inflammatory cytokines on prostate cancer phenotypic plasticity in vitro. We demonstrate that PC3ML metastatic prostate cancer cells exposed to EVs from adipose tissue ECs and to EVs from human adipose tissue total explants display reduced invasion and increased proliferation. The latter functional changes could be attributed to the EV miRNA cargo. We also show that the functional shift is *TWIST1*-dependent and is consistent with mesenchymal-to-epithelial transition, which is key to establishment of secondary tumor growth. Understanding the complex effects of EVs on prostate cancer cells of different phenotypes is key before their intended use as therapeutics.

## 1. Introduction

In the year 2020 alone, the American Cancer Society reported that 191,930 men were diagnosed with prostate cancer and that 33,330 men died from the disease [1]. While prostate cancer in general is relatively treatable, those who do develop metastatic disease have significantly higher mortality [2]. The prognosis is especially concerning for patients with other existing co-morbidities, especially obesity. Multiple studies have shown that the incidences of metastatic disease and poor outcomes are significantly greater for obese men [3,4]. Despite advances in cancer research, management of metastatic prostate cancer is clinically challenging with significantly poorer outcomes [2].

Tumors contain a heterogenous population of cell phenotypes including subpopulations shown to possess a high degree of metastatic potential. Prostate cancer metastasis preferentially targets the bones, including the vertebrae [5,6,7]. The original PC3 prostate cancer cell line was itself isolated from prostatic adenocarcinoma prostate cancer that was metastasized to the vertebrae of castration-resistant prostate cancer patients [8]. Highly metastatic PC3ML sublines were selected from PC3 parent cells for invasion and metastasis in vitro and in vivo [9]. Briefly, the cells were selected for invasion following three consecutive cycles using a transwell chamber with a barrier consisting of Matrigel reconstituted basement membrane material resulting in a subline with invasion four-to-five-fold higher than that of the parent cell line. Implantation of these cells into SCID mice resulted in secondary metastasis to various sites including lung and lumbar regions. Following five successive isolations followed by a one-step selection for invasion using the transwell chambers produced a highly invasive subpopulation with a high degree of metastatic potential and lumbar-targeting specificity [9].

In preparation for metastatic colonization, tumors can signal over long distances to promote the formation of the pre-metastatic niche, a hospitable environment that is favorable to tumor cell growth [10]. A hallmark of metastatic progression in cancer is the phenotypic transformation epithelial-to-mesenchymal transition (EMT). EMT occurs as phenotypically epithelial cells switch from apical–basal polarity to anterior–posterior polarity; their intercellular junctions become disrupted and permit increased migratory potential in the now phenotypically mesenchymal cells [11,12,13]. The mesenchymal phenotype is characterized by increased invasion and migratory abilities and has been linked with metastatic progression in human tumors. Expression of EMT markers is correlated to poor clinical prognosis. Although EMT is required for tumor cell dissemination, a less well understood process, mesenchymal-to-epithelial transition (MET) is vital for metastatic colonization. Indeed, it has been shown that tumor cell populations have a high degree of phenotypic heterogeneity with varying populations of epithelial, mesenchymal, and epithelial-mesenchymal hybrid cells, the distribution of which has been correlated with clinical prognosis [14,15,16]. Recent studies have shown that tumor cells that retain more epithelial features were uniquely capable of macrometastasis as compared to more mesenchymal cells in prostate cancer models [17]. Our findings in this paper support the contribution of previously unrecognized concerted contributors, such as extracellular vesicles produced by adipose tissue endothelium in pro-inflammatory environments and the EMT transcription factor *TWIST1*.

Inflammation has been identified as a biological mechanism linking cancer and obesity [18]. Patients with chronic inflammatory diseases are more predisposed to the development of cancer [19,20]. Abdominal obesity in particular is associated with an increased risk of aggressive prostate cancer [4]. The obese adipose environment is characterized by chronic inflammation driven in part by increased macrophage infiltration and elevated cytokine and adipokine expression [21]. This results in perturbed immune response and metabolic irregularities [22,23]. This chronic inflammatory state leads to chemoresistance, dysregulated angiogenesis, promotion of cancer cell growth, and the development of metastasis [22,23,24]. Obesity has been shown to alter the gene expression profiles of periprostatic adipose tissue, creating an environment that favors prostate cancer progression by promoting cancer cell proliferation and immune escape [25,26]. Another potential link between obesity and prostate cancer progression is endothelial-to-mesenchymal transition (EndMT), which has been observed in several diseases that are associated with inflammation including obesity [27,28,29]. EndMT is a maladaptive response whereby endothelial cells revert to a mesenchymal phenotype leading to impaired barrier function and angiogenic capacity. Our lab has previously shown that EndMT in the proinflammatory obese adipose microvascular environment produces extracellular vesicles (EV) that can induce dysfunction in naïve recipient endothelial cells [27].

The contribution of EVs to metastatic progression and the tumor microenvironment has been extensively investigated and summarized in a recent review by Ku et al. [28]. In summary, EVs have been shown to contain protein and nucleic acid cargo that participate in intercellular signaling and may promote tumor progression and establishment of the pre-metastatic niche [28]. EV-contained miRNA and protein have been found to promote invasion and metastasis in multiple cancer types [29,30,31,32,33,34].

*TWIST1* is a key driver of both EMT and EndMT. Regulation of *TWIST1* occurs through multiple different upstream regulators and is highly context dependent [35,36]. *TWIST1* may regulate inflammation in adipose tissue [37]. *TWIST1* also facilitates tumor invasion and metastasis by promoting EMT, formation of invadopodia, intravascular migration, extravasation, and vasculogenic mimicry (VM) [38,39,40,41]. *TWIST1* is highly expressed in prostate cancer, and expression levels are correlated with higher Gleason scores and poor prognosis [42]. *TWIST1* acts in concert with other EMT transcription factors including *SLUG* and *SNAIL*, and expression of *TWIST*, *SLUG*, and *SNAIL* is associated with poor outcomes in prostate cancer patients [43]. The biological effects exerted by *TWIST1* can occur through multiple downstream pathways. *TWIST1* can act either by regulating protein expression, as a transcriptional factor, or by modulating function [44]. *TWIST1* can form both homodimers and heterodimers with E2A or HAND proteins, and the ratio of homodimer to heterodimer functions as an important regulatory device.

We sought to explore whether the increased incidence of cancers in patients with SCI could be explained in part by EVs released by dysfunctional ECs from a proinflammatory environment and to determine the role of endogenous *TWIST1* on EV functional impact on recipient cells’ phenotype. Our lab has previously shown that EndMT occurs in the microvasculature of obese adipose tissue and that ECs in a proinflammatory environment release EVs that contain a proteomic signature that is distinct from the parent cell [27]. Here we demonstrate that HAMVEC EV internalization by PC3ML is cell cycle-dependent, and that the mechanism of internalization is independent of vesicle type. HAMVEC EVs from both a control (EV C) and a proinflammatory environment (EV PIC) reduced invasion, while only EV PIC increased proliferation. The transcriptome of EV C-and EV PIC-treated cells exhibited deregulation of genes consistent with the functional changes resulting from EV treatment. Increased endogenous *TWIST1* promoted both proliferation and invasion while deficient endogenous *TWIST1* resulted in reduced proliferation. In contrast to wild-type PC3ML, EV treatment did not induce any changes in proliferation or invasion in PC3ML deficient in *TWIST1*. EVs obtained from subcutaneous (SC ATEV) and omental (OM ATEV) adipose tissue of human subjects undergoing bariatric surgery exerted functional effects that mirrored those of EV PIC as well as increased glycolytic rate. SC ATEV contained a subset of miRNA that was also detected in HAMVEC EV. Transcriptomic analysis of PC3ML treated with SC ATEV compared to untreated revealed deregulation of genes mediating proliferation, invasion, and glycolysis. Collectively, these data suggest that EVs released from ECs in a proinflammatory environment may shift the malignant phenotype of recipient tumor cells and contribute to metastatic progression.

## 2. Results

### 2.1. EV Uptake and Cell Cycle

EV labeled with fluorescent DiD or dye only PBS controls were incubated with PC3ML overnight. Evaluation using Amnis ImageStream imaging flow cytometer revealed that double ultracentrifugation following DiD staining nearly eliminated DiD aggregate background fluorescence (Appendix A). EV internalization was assessed using IDEAS software internalization wizard and cells were sorted by cell cycle stage based on intensity of nuclear staining (Figure 1A). The majority of the PC3ML cell population, regardless of the EV treatment, were in the G0 cell cycle stage, followed by S and then G2/M. EV PIC-treated cells appeared to have a greater proportion of cells at G2/M phase than EV C-treated, 21.73% for EV PIC vs. 13.75% for EV C, although this was not statistically significant (Figure 1B). EV internalization was independent of vesicle type; internalization of EV C occurred at 58.22% and EV PIC at 60.55% (*n* = 5, *p* = 0.7396) (Figure 1C). We next evaluated the proportion of DiD-positive cells at each cell cycle stage. Both EV C and EV PIC were predominantly internalized during G0, followed by S phase, and the least EV uptake occurred at G2/M. Internalization was significantly different between each stage for both EV C- and EV PIC-treated cells (Figure 1D). To evaluate the mechanism of uptake, PC3MLs were treated with endocytosis inhibitors dynasore, genistein, fillipin, or nystatin prior to addition of labeled EVs. In both EV C- and EV PIC-treated cells, EV uptake was significantly reduced by treatment with Dynasore (EV C *p* = 0.0001, EV PIC *p* < 0.0001), Genistein (EV C *p* = 0.0004, EV PIC *p* < 0.0001), and Fillipin (EV C *p* = 0.021, EV PIC *p* = 0.0053) (Figure 1E). Treatment with endocytosis inhibitors did not change the relative proportions of cells in each stage of the cell cycle (Appendix A). Inhibitor treatment did however change the proportion of EV internalization positive cells at G0, and S in EV C- and EV PIC-treated cells. The clathrin-dependent endocytosis inhibitor dynasore and caveolin-dependent endocytosis inhibitor genistein significantly reduced the % uptake at G0 in EV C- and EV PIC-treated cells. (Figure 1E). This data shows that irrespective of EV source, internalization of EVs by PC3ML cells occurs in the G0 primarily via receptor-mediated endocytosis by both clathrin- and caveolin-dependent mechanisms. To determine the co-localization with lysosomes following uptake, we evaluated the percent of internalized EV colocalization with the lysosome. We found that only a small percentage of EVs were contained within the lysosome, with less than 20% colocalization after 24 h incubation with EV (Appendix A).

Next, we compared gene expression of PC3ML cells that were incubated with either EV C or EV PIC with control cells in absence of EV treatment. IPA was used to identify genes with ontology terms matching endocytosis, clathrin-mediated endocytosis, caveolae mediated endocytosis, or macropinocytosis. We found that EV C downregulated four and upregulated seven genes involved in endocytosis, as compared to controls, out of which three mediated clathrin-mediated endocytosis, and two of the latter also mediated caveolae-mediated endocytosis; *IFNG* was also upregulated and is known be involved in macropinocytosis (Figure 1F).; EV PIC upregulated eight and downregulated five genes involved in endocytosis. *APOD* was the only upregulated gene known to mediate clathrin-dependent endocytosis and two upregulated and one downregulated genes mediated macropinocytosis (Figure 1F). EV miRNA cargo was assessed using the NanoString miRNA array; 16 miRNA were detected in both EV C and EV PIC and four only in EV PIC (Table 1). IPA was used to identify experimentally validated target genes of detected miRNA (Appendix A). We identified *PLAU* as being downregulated in both EV C- and EV PIC-treated cells and as a target of miR-23, which was detected in both vesicle types. Endocytic regulation of *PLAU*/*PLAUR* may serve to regulate a cell’s migration and invasion capacity and has been implicated in functions including EMT in several cancer types [45,46,47,48].

### 2.2. EV C and EV PIC Alter Functional and Molecular Signature of PC3ML

Invasion and proliferation are two key functional features that, together, can inform on the predominant phenotype of malignant cells. We assessed both functions in the PC3ML cells following exposure to EV C or EV PIC compared to untreated cells. The impact of HAMVEC EV on PC3ML invasion was determined by seeding EV-treated PC3ML onto a 3D Matrigel matrix. Invasion was reduced in both EV C- (*p* = 0.0147) and EV PIC- (*p* = 0.0405) treated cells (Figure 2A). PC3MLs were incubated overnight with EV C or EV PIC, and proliferation was measured using fluorescent microscopy to assess BrdU incorporation. EV PIC but not EV C increased proliferation by 1.3-fold (*p* = 0.0005) (Figure 2B). This dual effect is consistent with the phenotypic shift that occurs during mesenchymal-to-epithelial transition (MET); the cells shift from a less proliferative, more invasive phenotype to a more proliferative less invasive phenotype, which is necessary for the colonization of secondary tumor sites.

Transcriptomic analysis revealed that 38 genes were differentially expressed (DE) +/− 0.58 log2 fold change, *p* < 0.05, in PC3MLs treated with EV C as compared to untreated, and 70 genes were DE in PC3MLs treated with EV PIC vs. untreated (Figure 3A, Table 2 and Table 3). IPA was used to obtain a list of 415 genes with the ontology category of “invasion of tumor”. This list was then compared against the DE genes in EV C and EV PIC vs. untreated PC3ML and against the experimentally validated mRNA targets of the miRNA contained within EV C and EV PIC. The latter are reported in Appendix A. Three genes whose downregulation is known to inhibit invasion (*CDK2N2A*, *PLAU*, *ITGB1*) were DE in EV C vs. untreated PC3ML, and six genes whose regulation status is known to inhibit invasion (*CD82*, *COL7A1*, *ENPP2*, *KISS1*, *PLAU*, *PLAUR*) were DE in EV PIC vs. untreated PC3ML. *PLAU* was downregulated in both EV C- and EV PIC-treated cells and is a validated target gene of miR-23a, which was detected in both EV C and EV PIC (Figure 3B). 55 genes were differentially expressed +/− 0.58 log2-fold change in PC3MLs treated with EV PIC as compared to PC3MLs treated with EV C (Figure 3C, Table 4). IPA was used to obtain the list of 1163 genes with the ontology category “proliferation of tumor cells”. This list was then compared against the DE genes in EV PIC- vs. EV C-treated PC3MLs and against the experimentally validated mRNA targets of miRNA contained within EV C and EV PIC. 13 DE genes were found to be associated with proliferation with a regulation status consistent with increased proliferation (Figure 3D). In order to obtain a visual reference for transcriptomic changes and resulting malignant signature of PC3ML treated with EV C or EV PIC, DE genes were sorted by annotations provided with the NanoString PanCancer Progression panel and clustered by their involvement in extracellular matrix remodeling, metastasis, and epithelial-mesenchymal transition (EMT). EV C treatment resulted in deregulation of 16 genes mediating extracellular matrix remodeling, 10 genes mediating metastasis, and 10 genes mediating EMT (Figure 3E). EV PIC treatment resulted in deregulation of 25 genes mediating extracellular matrix remodeling, 12 genes mediating metastasis, and 23 genes mediating EMT (Figure 3F). While EVs released from ECs in a proinflammatory environment and control environment impacted on PC3ML invasion and transcriptome, only EVs released from the proinflammatory ECs increased proliferation in addition to a marked upregulation of genes involved in EMT. This indicates that the proinflammatory EC environment may produce EVs that confer some phenotypic flexibility in part by stimulating EMT signaling pathways.

### 2.3. Endogenous Twist1 Expression Levels Change Functional and Molecular Phenotype of PC3ML

High expression levels of *TWIST1* and *SNAI1* are known hallmarks of PC3ML and are positively correlated with Gleason scores [42,49,50]. PC3MLs deficient in *TWIST1* (ΔT) were generated using a single allele deletion; cells overexpressing *TWIST1* (T^oe^) were generated using a constitutive expression vector driven by *EL1α1*. Proliferation was impacted by endogenous *TWIST1* expression; cells deficient in *TWIST1* showed significantly reduced proliferation (*p* = 0.0039) while overexpression of *TWIST1* significantly increased proliferation (*p* = 0.0032), as compared to wild-type (WT) cells (Figure 4A). In addition, invasion capacity of PC3ML cells was changed based on endogenous *TWIST1* expression. Overexpression of *TWIST1* significantly increased invasion as compared to WT (*p* = 0.0278) while *TWIST1* deficiency had no effect (Figure 4B). Differential expression analysis of the ΔT and T^oe^ transcriptomes as compared to WT revealed 12 deregulated genes in ΔT and 29 genes deregulated in T^oe^ as compared to WT (Figure 4C, Table 5 and Table 6). Deregulated genes were then compared against a curated list of genes mediating proliferation and invasion as previously described. Deficient *TWIST1* expression resulted in deregulation of five genes associated with proliferation (*CXCL8*, *EDN1*, *F3*, *PLAU*, and *RB1*) in a direction consistent with observed reduction in proliferation. Overexpression of *Twist1* deregulated the expression of three genes associated with proliferation (*DICER1*, *PIK3CA*, and *TWIST1*) in a direction consistent with observed increases in proliferation (Figure 4D). Low *TWIST1* expression deregulated four genes associated with invasion (*EDN1*, *F3*, *PLAU*, and *SYK*) in a direction that is consistent with reduced invasion; however, this was not observed in the functional assay. Overexpression of *TWIST1* upregulated two genes associated with invasion (*DICER1* and *TWIST1*) in a manner consistent with the observed increase in invasion (Figure 4E). *TWIST1* overexpression promotes both invasion and proliferation in the T^oe^ cells as compared to WT PC3ML. This suggests that *TWIST1* overexpression provides the cells with the phenotypic flexibility to adapt to stimuli encountered in the local environment while single allele knockdown may have resulted in a hybrid epithelial-mesenchymal phenotype.

These data demonstrate that PC3ML proliferation is correlated to *TWIST1* expression and that while reduced *TWIST1* expression does not affect PC3ML invasion, increased expression of *TWIST1* significantly increases invasion. Variation in *TWIST1* expression alters expression of genes mediating proliferation and invasion in a manner consistent with observed functional effects. By altering *TWIST1* endogenous expression, we expanded the range of phenotypes for PC3ML cells, thereby providing a cellular tool to refine the impact of EVs based on multiple malignant phenotypes.

### 2.4. PC3ML Endogenous Twist1 Levels Alter the Functional and Molecular Effect of Endothelial EV

Unlike the PC3ML wild-type cells, PC3MLs deficient in *TWIST1* did not show any changes in proliferation or invasion following EV treatment (Figure 5A,C). However, PC3MLs overexpressing *TWIST1* showed increased proliferation in response to EV PIC (*p* = 0.0434) and reduced invasion in response to EV C (*p* = 0.0095) and EV PIC (*p* = 0.0125) (Figure 5B,D), similarly to functional changes displayed by the wild-type cells. The transcriptome of EV PIC-treated ΔT and T^oe^ was compared to that of EV PIC-treated WT cells. 75 genes were found to be deregulated +/− 0.58 log2 fold change in ΔT treated with EV PIC as compared to WT treated with EV PIC, and 35 genes were found to be deregulated +/− 0.58 log2 fold change in T^oe^ treated with EV PIC as compared to WT treated with EV PIC (Figure 5E, Table 7 and Table 8). Deregulated genes were then evaluated for their potential association with proliferation or invasion. Evaluation of genes associated with proliferation revealed that 11 genes were deregulated in EV PIC-treated ΔT and five genes in EV PIC-treated T^oe^ as compared to EV PIC-treated WT (Figure 5F). Evaluation of genes associated with invasion revealed that 11 genes were deregulated in EV PIC-treated ΔT and four genes in EV PIC-treated T^oe^ as compared to EV PIC-treated WT (Figure 5G). These results suggest that variation in expression levels of endogenous *TWIST1* has a significant functional impact on the response of the recipient cells with EV treatment. Low endogenous levels of *TWIST1* completely blunt the effect of EVs on proliferation and invasion, while EVs’ effect on proliferation and invasion in high-expressing *TWIST1* cells is similar to effects on PC3ML wild-type controls. This finding emphasized the importance of the sub-phenotypes within a highly metastatic cell population that must be taken into consideration when EV therapies are aimed to induce targeted functional effects.

### 2.5. Adipose Tissue Extracellular Vesicles Induce Functional Changes in PC3ML

Obesity is a source of SCI and is associated with an increased incidence of multiple cancer types. Since abdominal obesity is implicated in a large number of abdominal cancers, including prostate cancer, we sought to determine the contribution of both omental (OM) and subcutaneous (SC) adipose tissue EVs on prostate cancer cell phenotype. EVs were isolated from adipose tissue explants of paired OM and SC depots, as described under methods. Adipose tissue (AT) EVs’ concentration, size, and morphology were determined for each preparation (Appendix A). In addition, EV markers were confirmed in all of the EV preparations using LC/MS/MS (Appendix A). PC3MLs were incubated overnight with either OM or SC ATEV, and proliferation was assessed using fluorescent microscopy analysis of BrdU uptake. Both SC and OM ATEV increased proliferation (*p* = 0.0064, *p* = 0.0431) as compared to untreated controls (Figure 6A). The impact of ATEV on the invasive phenotype of PC3ML was determined by seeding EV-treated PC3MLs onto a 3D Matrigel matrix. The PC3ML cells that received SC and OM ATEVs exhibited significant reduction in invasion (*p* < 0.01, *p* < 0.01) (Figure 6B). To determine if ATEV changed the metabolic phenotype of PC3ML, glycolytic rate of untreated and SC- or OM-treated cells was measured using the Agilent Seahorse metabolic analyzer. Both SC and OM ATEV increased the glycolytic rate of recipient PC3ML (*p* < 0.01, *p* < 0.05) (Figure 6C). NanoString miRNA array was used to evaluate the miRNA cargo contained in ATEV. Three biological replicates were assessed for each vesicle type, and 48 miRNA were detected in SC ATEV, 55 in OM ATEV, and 44 were detected in both OM and SC (Table 9). Nine miRNA detected in SC and OM ATEV were also found in EV C and EV PIC (Table 10). IPA was used to identify experimentally validated target genes of detected miRNA (Appendix A). Next, transcriptomic analysis was performed on PC3ML cells before and after treatment with ATEVs, which revealed that 108 genes were DE +/− 0.58 log2 fold change in ATEV-treated vs. untreated PC3ML (Figure 7A, Table 11). IPA was used to generate lists of genes with the ontology categories of “glycolysis”, “proliferation”, or “invasion”. These lists were then compared against the DE genes found in SC ATEV-treated compared to untreated control PC3ML cells. ATEV treatment resulted in deregulation of 29 genes associated with proliferation, 17 genes associated with invasion, and 17 genes associated with glycolysis (Figure 7B). Seven genes associated with glycolysis had an activation state consistent with observed functional increase in glycolytic rate in ATEV-treated cells (Figure 7C). Five genes that were downregulated in SC ATEV-treated cells had an activation state consistent with reduced invasion observed in functional assay and were identified as experimentally validated target genes of miRNA detected in ATEV (Figure 7D). Four genes that were downregulated in SC ATEV-treated cells had an activation state consistent with both PC3ML increased proliferation observed in functional assay and were identified as experimentally validated target genes of miRNA detected in ATEV (Figure 7E). Collectively, this data shows that EVs from both subcutaneous and omental fat depots increase proliferation and glycolytic capacity and reduce invasion of PC3ML cells. Consistent with the functional data, we also found that miRNA contained within ATEV had validated target genes mediating proliferation and invasion, and the latter were also DE following ATEV treatment.

## 3. Discussion

Our data shows that EV internalization by prostate cancer cells is an active and indiscriminate process driven by primarily clathrin-dependent endocytosis in a cell cycle-dependent manner. Internalization occurs at G0, potentially in migratory cells. Previous literature shows that clathrin-mediated endocytosis is shut down in cells that actively proliferate [51]. However, this finding was never purported to explain the EV uptake in malignant cell phenotypes. Although receptor-mediated endocytosis was found as the predominant mechanism of EV uptake, other mechanisms, such as macropinocytosis, also have a contribution. Malignant cells are known to enhance classic and non-canonical pathways of uptake that support their plasticity [52]. Often, the uptake of EVs based on cellular phenotype is overlooked, and therefore the therapeutic approaches that use EVs as delivery systems may have limited efficacy. The implications of our finding emphasizes the concept that targeting proliferating cells using EV approaches in cells that primarily engage receptor-mediated endocytosis may be challenging, and additional functionalized approaches may be needed to make such targeting effective. The fate of the EVs once internalized is also of prime importance as it impacts on the delivery location of the cargo. We show that only a small proportion of the EVs are engaging in a lysosomal route. Recent findings show that the lysosomal destination of EVs is key for release of their cargo in the cytosol [53,54,55]. We discovered a significant degree of overlap in EV miRNA cargo despite various EV sources. There is evidence to suggest that EV targets and downstream effects may be influenced by factors including tetraspanin composition, at the EV surface [56,57]. A recent study has found evidence of a functional protein corona surrounding EVs that influences the functional effect of those EVs on recipient cells [58]. The divergent effects of vesicles with similar cargo suggests that the fate of EV cargo and its downstream influence on cell function may be vesicle-dependent, and future investigations will be key to support this hypothesis.

Both EV C and EV PIC reduce invasion, but only EV PIC increases proliferation. EV C and EV PIC inhibition of invasion despite upregulation of MMPs may be explained by dysregulation of key extracellular proteolytic mediators including *PLAU* and *PLAUR*. *PLAU* and *PLAUR*, which were downregulated with EV treatment, are critical pro-MMP activators [59]. While upregulation of *PLAU* is associated with upregulation of MMP activity and promotion of tumor cell invasion, the observed downregulation of these genes may explain the lack of invasion in the presence of upregulated MMP [60]. Additionally, *CD82*, which was upregulated in EV PIC-treated PC3ML, had been shown to suppress invasion by inactivating *MMP9* [61,62].

This implies that EV PIC is providing additional plasticity to PC3ML cells to adapt to environment, which is associated with some of the most aggressive and treatment-resistant cancers [63,64]. Several miRNAs were detected in EV PIC but not EV C, which may account for some of the observed functional effects and have implications for future therapeutic strategies. MiR-128 overexpression, found in the EV-PIC, has been shown to shift cells to a more epithelial phenotype, increased protein expression of E-cadherin, and reduced protein expression of N-cadherin in pancreatic cancer cells [65]. This shift to a more epithelial phenotype may speak to increased phenotypic plasticity, and the shift away from mesenchymal transition is consistent with a more proliferative phenotype. MiR-155 dysregulation has been observed in numerous malignancies [66,67]. Another miRNA found in EV PIC and ATEV, miR-155, has been shown to promote proliferation in prostate cancer cells; expression of miR-155 is elevated in prostate tumor tissue as compared to para-carcinoma tissue and was found to be positively correlated to tumor volume and metastasis [68,69]. Development of a co-expression and competitive endogenous RNA network of cirRNAs that are significantly deregulated in prostate cancer samples revealed that the EMT transcription factor *SMAD4* may be partially regulated by miR-1285 (which was detected in EV PIC); additionally, hsa-cir-0001206 binding of miR-1285 inhibits prostate cancer cell proliferation, migration, and invasion [70]. These miRNAs detected only in EV PIC all mediate some aspect of EMT and collectively may impact on the epithelial/mesenchymal phenotype of prostate cancer cells. Therefore, horizontal delivery of such miRNAs via EVs may aid in phenotypic plasticity of metastatic prostate cancer cells. Multiple glycolytic enzymes were upregulated in ATEV-treated PC3ML, consistent with observed increases in glycolytic rate. EVs have been implicated in enzyme transfer and subsequent metabolic reprograming in numerous cancer types [71,72,73]. Systemic inflammation has known associations with cancer incidence and aggressiveness, and our in vitro system that mimics chronic inflammation impacts on prostate cancer cells’ plasticity via EVs produced by either endothelial cells from adipose tissue or EVs collectively produced by multiple cells that reside in adipose tissue, collected from patients with obesity [19,20].

Patients with chronic inflammatory diseases are at an increased risk of developing cancer. Both diabetes mellitus and obesity (particularly abdominal obesity) are associated with systemic chronic inflammation and with intermediate- and high-risk prostate cancer [3,4,74,75]. *TWIST1* is a key mediator of both EndMT and EMT and is highly expressed in both obesity and prostate cancer. Because of the known involvement of *TWIST1* and its facilitation of invasion and metastasis, we generated PC3ML cells with various expressional levels of endogenous *TWIST1* and interrogated the impact on PC3ML response to EVs. We found that low endogenous *TWIST1* resulted in a loss of effect of EVs on cell proliferation and invasion. *TWIST1* is a validated target of some of the miRNA found in the EV PIC and ATEVs. Notably, miR21 and miR-23 are known to target *TWIST1* and impact on EMT. Mir-21 is an oncogene, and expression of miR-21 is associated with upregulation of *Twist1* [76]. Expression of miR-21 has been shown to increase the transcriptional activity of *TWIST1* and associated genes in multiple cancer types [77,78,79]. Expression of miR-23 is correlated with *TWIST1* expression and EMT in several malignancies [80,81,82]. Aberrant expression of EMT transcription factors has been associated with drug resistance in some cancers [82]. Conversely, drug sensitivity has been shown to increase upon deletion of *TWIST* in a transgenic mouse model [83,84]. Since cells with only one *TWIST* allele may have a minimal response to miRNA that target the *TWIST* gene, this may support in part the lack of functional effects as a result of treatment with EVs that contain such miRNAs as their cargo.

Collectively, these data suggest that EVs from a proinflammatory environment may exacerbate the malignant phenotype of prostate cancer cells by targeting the *TWIST1*/EMT signaling axis and that this axis may serve as a potential therapeutic target in highly metastatic cells.

The prospect of utilizing nanomedicine technology for targeted drug delivery with minimal off-target toxicity has garnered much interest in recent years. Unfortunately, the efficacy of such therapies has of yet not met the threshold required for practical clinical implementation. Numerous nanomaterials have been tested as agents for chemotherapeutic delivery including lipid nanoparticles and liposomes, metallic nanoparticles, micelles, and polypeptides [85,86,87,88,89,90,91]. Barriers include first mass metabolism and the fact that the nanoparticles do not passively migrate through “leaky” gap junctions, but instead require active transcytosis to reach the solid tumor [92]. Our findings have implications for future approaches of therapies that are aiming to use EVs as a functionalized and targeted delivery system. The apparent active nature of nanoparticle delivery suggests that strategic targeting of these particles is required for optimal delivery to solid tumors. Interestingly, NDRG1, one of the most upregulated genes in ATEV-treated PC3ML, is known to be involved in vesicular recycling (specifically of E-cadherin) via Rab4 [93]. We found that PC3ML cell EV uptake is indiscriminate and that G0, possibly migratory cells, are preferentially internalizing EVs. This suggests that these cells may be more effective candidates for targeted delivery of chemotherapeutics.

Epithelial-mesenchymal plasticity and heterogeneity of the epithelial-mesenchymal state of cells within a tumor have been cited as contributors to multidrug resistance in cancer [94]. Overexpression of the EMT TFs *SNAIL* and *SLUG* in lung cancer cell lines are associated with resistance to gefitinib [82]. The current approved therapies for advanced, metastatic castration-resistant prostate cancer include androgen-receptor axis-targeted agents, taxane chemotherapy, radium-223, and sipuleucel-T [95]. These treatments are unable to completely eliminate the tumors however and only slow metastatic progression. Prostate cancer tumors are immunosuppressive and therefore unresponsive to current immunotherapies [96]. Deletion of *TWIST* in a transgenic mouse model was shown to increase sensitivity to gemcitabine [83,84]. The proinflammatory HAMVEC EV PIC exacerbated the malignant phenotype of PC3ML by increasing proliferation and deregulating genes mediating metastatic processes including metastasis and EMT. Overexpression of *TWIST1* increased both proliferation and invasion in PC3MLs suggesting enhanced phenotypic flexibility that may support enhanced ability of cells to rapidly adapt to a variety of environmental conditions. Single allele deletion of *TWIST1* in PC3ML eliminated any EV C- or EV PIC-induced changes in cell proliferation or invasion. ATEV similarly enhanced proliferation and inhibited invasion in PC3MLs while also enhancing glycolytic capacity. Collectively, these data suggest that EV from a proinflammatory environment may exacerbate the malignant phenotype of prostate cancer cells by manipulating the *TWIST1*/EMT signaling axis and that this axis may serve as a potential therapeutic target.

Interestingly, notwithstanding the population-level evidence that appears to support a straightforward link between obesity and prostate cancer, emerging evidence suggests a much more nuanced relationship. BMI correlates with more aggressive prostate cancer; however, patients with BMI greater than 30 actually have better outcomes [97,98]. Evidence regarding more advanced, metastatic castration-resistant prostate cancer is complex with data both supporting and refuting a correlation between BMI and prostate cancer risk [98,99,100,101,102]. A meta-analysis by Harrison et al. shows no correlation between BMI and prostate cancer risk and only weak evidence for reduced risk with obesity [103]. In prostate cancer patients with advanced disease and failure of both androgen deprivation and withdrawal, obesity was associated with better overall survival and reduced cancer-specific mortality [98,99,104]. While much attention has been paid to the influence of omental adipose tissue and its relationship to systemic obesity, there is evidence of the contribution of subcutaneous adipose tissue to prostate cancer progression. Both visceral and subcutaneous adipose tissues have been shown to be associated with aggressive pathological features in prostate cancer [105,106,107]. EVs isolated from subcutaneous adipose tissue have been demonstrated to increase melanoma aggressiveness and were found to contain proteins that are associated with fatty acid oxidation, and there is evidence of cross-talk between peripheral adipose tissue, periprostatic carcinoma adipose tissue, and prostate cancer progression [26,107,108]. A greater understanding of the contributions of both omental and subcutaneous adipose tissue depots may inform on the seemingly paradoxical relationship between obesity and prostate cancer progression and metastasis.

One limitation of this study is that experiments were performed only on highly metastatic PC3MLs. The effects of adipose EVs on PC3 and DU145 cells as well as the less aggressive 22RV21 cells have been well studied, but the molecular underpinnings of metastatic prostate cancer are less well understood [26,109,110,111,112,113]. We sought to fill this gap in knowledge by exploring the mechanisms driving more advanced and aggressive disease. Another limitation is that the ATEV used in these experiments were obtained from both diabetic and non-diabetic subjects. While the results between both subject types appear consistent, future studies should examine this relationship in further detail.

## 4. Materials and Methods

### 4.1. Human Subjects

For all studies involving human subjects, informed consent was obtained, and the Eastern Virginia Medical School Institutional Review Board approved the research project. The study included a cross-sectional cohort of morbidly obese type 2 diabetic (T2D) and non-diabetic subjects, aged 18–65 years, undergoing bariatric surgery at Sentara Metabolic and Weight Loss Surgery Center (Sentara Medical Group, Norfolk, VA, USA). Exclusion criteria included autoimmune disease, including type 1 diabetes mellitus, conditions requiring chronic immunosuppressive therapy, anti-inflammatory medications, thiazolinendiones, active tobacco use, chronic or acute infections, or a history of malignancy treated within the last 12 months. T2D was defined as a fasting plasma glucose of 126 mg/dL or greater, a glucose of 200-mg/dL or greater after a 2.0-h glucose tolerance test, or use of antidiabetic medications.

### 4.2. Adipose Extracellular Vesicle Isolation

Paired omental (OM) and subcutaneous (SC) adipose tissue (AT) was collected from human bariatric subjects undergoing surgery at Sentara’s Surgical Weight Loss Center. Approximately 3.0 g aliquots of OM and SC AT were transferred to a clean, sterile scintillation vial. An amount of 5 mL of 2%FBS/DMEM/F12 (1:1) + 1% penicillin/streptomycin was added to the vials, and the AT was minced with scissors. The contents of the vials were transferred to a petri dish and supplemented with 10 mL of media. The OM and SC petri dishes were allowed to incubate for 18-24 h. After incubation, the media from both depots were filtered through a 70 μm cell strainer to remove the small AT pieces. The sample was then centrifuged at 500× *g* for 10 min to remove cells, and the supernatant was transferred and centrifuged at 10,000× *g* for 40 min to remove cellular material and protein. The supernatant was transferred again, and EVs were pelleted for isolation by centrifuging samples twice, at 100,000× *g* for 90 min. The EV pellet was resuspended in 500 µL of PBS. To determine particle concentration and size distribution, particles were diluted 1:100 in PBS, and nanoparticle tracking analysis was performed using the NanoSight 300 (Camera Level: 12-15, Screen Gain: 1, Capture Number: 3, Capture Time Length: 30-s, Temperature: 25 °C, Detection Threshold: 5). Vesicles were imaged using electron microscopy, and vesicle markers were characterized via LC-MS/MS mass spectrometry as previously described [27]. Both OM and SC ATEV were verified to contain CD9, CD63, CD81, Syntenin-1, and ALIX (Appendix A).

### 4.3. Endothelial Cells

Human adipose microvascular endothelial cells (HAMVEC) were purchased from Sciencell Research Laboratories (cat#:7200). HAMVEC were cultured on fibronectin-coated plates using endothelial cell medium complete kit (Sciencell, cat# 1001) in a 37 °C, 5% CO_2_ incubator. HAMVEC experiments were conducted between passages 4-8. HAMVEC were stimulated with 5.0 ng/mL of TGFβ, IFNγ, and TNFα for 6 days of pro-inflammatory cytokine (PIC) treatment.

### 4.4. Endothelial Cell Extracellular Vesicle Isolation and Characterization

Cell culture media from control and PIC-treated HAMVEC were collected and centrifuged at 500× *g* for 10 min to remove dead cells, and supernatant collected. EVs isolated from HAMVEC receiving PIC treatment are referred to as EV PIC, and those from untreated control HAMVEC are termed EV C. The sample was subsequently processed and characterized as described above for the adipose tissue EVs and in previously published research [27].

### 4.5. PC3ML Metastatic Prostate Cancer Cells

Immortalized metastatic prostate cancer cells (PC-3ML) were obtained from our collaborator, Dr. Oliver J. Semmes’ group from the Leroy T. Canoles Cancer Research Center, at Eastern Virginia Medical School. PC3-ML cells were grown in 10% FBS/DMEM/F12 (1:1) media + 1% penicillin/streptomycin (Thermo Fisher, cat#: 11320033) in a 37 °C, 5% CO_2_ incubator.

### 4.6. Generation of PC3ML Deficient in Twist1 by CrisprCas9

A two-guide strategy was employed to maximize the likelihood of gene knockout. Two gRNAs, targeting distinct exons, were introduced to the cell to generate a double-stranded (ds) DNA break, and a *Twist1* homology blasticidin (Bsd) selection resistance gene was inserted in place of the excised gene. The vector containing the first gRNA, targeting a cut within the 11th codon of Twist1, and Cas9 nickase was generated in an *E. coli* stbl3 host. The vector containing the second gRNA, targeting a cut within the 18th codon of Twist1, allows for efficient insertion and deletion and was generated in an *E. coli* stbl3 host. The homology vector for insertion of Bsd resistance selection was generated in an *E. coli* stbl3 host. Vector isolation was performed using a ZymoPure plasmid midiprep kit (Zymoresearch, cat # D4213). Plasmids were linearized using BsiWI restriction enzyme (New England BioLabs, cat # R0553S). PC3-ML were transfected with all three plasmids using Nucleofector^TM^ (Lonza, Cat # VPB-1003). Selection was performed using 5 μg/mL blasticidin (Gibco, Cat # R21001).

### 4.7. Generation of PC3ML Cells Overexpressing Twist1

PC3-MLs overexpressing Twist were generated by insertion of a piggyback transposon vector that drives constitutive *Twist1* expression with the human eukaryotic translation elongation factor 1α1 promoter. This vector contains the selection resistance gene for geneticin and was generated in an *E. coli* Stbl3 host. Vector isolation was performed using a ZymoPure plasmid midiprep kit (Zymoresearch, Irvine, CA, USA, cat # D4213). Plasmids were linearized using BsiWI restriction enzyme (New England BioLabs, Ipswich, MA, USA, cat # R0553S). PC3-MLs were transfected using XFECT^TM^ transfection reagent (Takara Bio, Kusatsu, Japan, Cat # 631317). Selection was performed using 500 μg/mL Geneticin (G418) (Life Technologies Corp, Carlsbad, CA, USA, Cat # 10131035).

### 4.8. Extracellular Vesicle Labeling

Vibrant DiD label (Thermo Fisher Scientific, cat # V22887) was diluted 1:200 with PBS containing EVs or PBS alone as a dye control and incubated for 10–20 min at 37 °C. Samples were incubated for 10–20 min at 37 °C; then excess dye was removed using two rounds of ultracentrifugation at 100,000× *g* for 90 min. Samples were used immediately or stored up to 48 h at 4 °C protected from light.

### 4.9. Extracellular Vesicle Uptake and Cell Cycle Analysis

PC3ML (250,000 cells) were seeded onto 6-well plates and grown until approximately 70% confluency. Cells were then incubated overnight with 100,000 EVs/cell of DiD-labeled EV C or EV PIC derived from control or PIC-treated HAMVEC. The media was removed, and wells were washed with PBS. Cells were then trypsinized and pelleted at 220× *g* for 5 min. The supernatant was discarded, and cell pellets were washed in 1 mL of PEB Buffer (1X PBS, 0.5% BSA, 2 mM EDTA, pH 7.4). Cells were then fixed on ice in 2% formaldehyde/PBS for 20 min protected from light. After fixation, the cells were washed in 1 mL PEB, resuspended in 1:1000 DAPI in PBS (Invitrogen, Waltham, MA, USA, Cat # D1306), and incubated for 5 min. Cells were washed with 1 mL PEB and resuspended in 30μL 2% Fetal Bovine Serum in PBS (FBS, Sciencell cat no. #050, Carlsbad, CA, USA). Cells were analyzed on an AMNIS ImageStream Mark II instrument and acquisition, and analyses were completed using Ideas 6.2 software. Specifically, the Ideas internalization wizard was utilized to determine the percentage of cells in each population that internalized EVs. To do so, the wizard uses a mask feature to calculate the intensity of DiD-stained EVs found within the masks placed over the brightfield image cells. Effect of endocytosis inhibition was assessed by treating cells with 100 μM Dynasore (abcam, Cat #Ab120192), 100 μM Genistein (abcam, cat # Ab120112), 2 μg/mL Fillipin (Sigma, cat# F9765), or 25 μg/mL Nystatin (Sigma, St. Louis, MI, USA, cat # N6261) for one hour prior to addition of EV. Labeled EV solution or DiD in PBS control was then added to treated cells and incubated for an additional 4 h at 37 °C. Media containing endocytosis inhibitors and EVs was then aspirated, and cells were collected, fixed, and incubated with DAPI as described above. Co-localization was assessed by staining the lysosomes with LysoTracker^TM^ green DND-26 at a final concentration of 60 nM, 30 min prior to collection. Cells were imaged using the Amnis ImageStream, and EV uptake was assessed by analyzing internalization on IDEAS 6.2.187.0 software (Luminex, Austin, TX, USA), and colocalization was assessed using the colocalization wizard. Cell cycle phase was determined by gating cell population by DAPI intensity, frequency, and area. Cells were sorted into G0, S, or G2M based on relative intensity and area clustering of DAPI staining [114,115,116].

### 4.10. NanoString miRNA and RNA Transcriptome Analysis

Cellular RNA was isolated using the Direct-zol miniprep kit (Zymo Research Corporation, Cat # R2072). Exosomal RNA was isolated using a single-cell RNA purification kit (Norgen Bioteck Corp, Thorold, ON, Canada, Cat # 51800) or the GenElute single-cell RNA purification kit (Sigma Life Science, Cat # RNB300). The mRNA transcriptome of PCa cells was probed using the nCounter^®^ Pan Cancer Progression Panel (NanoString, Seattle, WA, USA, Cat # XT-CSO-PROG1-12). EV miRNA cargo was probed using the nCounter^®^ Human V3 miRNA assay (NanoString, Cat # CSO-MIR3-12). Raw data were quality control assessed and normalized by the NanoString nSolver^TM^ analysis and Rosalind Bio software version 3.35.22.1. Rosalind quality control of NanoString miRNA data was performed by assessing total number of fields of view captured and binding density of barcode capture by spots per square micron. The noise threshold was set by calculating the number of standard deviations between the average negative control value and the positive control. Ligation control threshold was calculated by determining the number of standard deviations of the positive controls that are above the negative controls. Normalization of miRNA data was performed by Rosalind software. Normalization consisted of a two-step data transformation involving a positive control r and codeset normalization factor. Both normalization factors were generated by calculating the geometric mean of the selected probes then the arithmetic means of those geometric means for all samples. The normalization factor, the ratio of the arithmetic mean vs. the geometric mean, was then multiplied by the counts for every probe by its lane-specific normalization factor. Ingenuity Pathway Analysis (Qiagen bioinformatics, IPA, Hilden, Germany) was used for network and miRNA target-gene enrichment analysis. Putative miRNA targets were predicted using the IPA miRNA target filter tool. IPA diseases and functions were also used to generate gene lists based on ontology functions relevant to functional assays.

### 4.11. PC3ML Cell Proliferation

5 × 10^4^ cells were seeded into 4-well chambered slides and allowed to grow up to 50–75% confluency. PC3ML were treated with EV PIC or EV C for 16–18 h. PC3ML were then incubated with 10 µM BrdU (Abcam, Cambridge, UK, cat#: ab142567) for 6 h. BrdU labeling solution was removed from the wells, and cells were washed three times with PBS. PC3ML were fixed with 2% formaldehyde/PBS at room temperature for 30 min. After three PBS washes, the cells were permeabilized with 0.1% Triton X-100 for 30 min, washed again, and treated with 2M hydrochloric acid (HCl) for 20 min, at 37 °C. HCl was removed and sodium tetraborate (0.1 M, pH = 8.5) was added for 30 min at room temperature. Cells were blocked with 10% NGS for 1 h, followed by incubation with anti-BrdU AlexaFluor 488 antibody (1:200; Santa-Cruz Biotechnologies, Dallas, TX, USA, cat#: SC-32323 AF488) and 5 μg/mL DAPI prepared in 10% NGS. Wells were cover slipped with Fluoromount G. Images were taken with an Olympus BX50 fluorescence microscope.

### 4.12. PC3ML Invasion Assay

3 × 10^5^ PC3-ML cells were treated for 16-18 h with 3 × 10^10^ OM and SC adipose tissue EVs isolated from three different human subjects. 10^5^ cells in 0.5% BSA/DMEM/F12 (1:1) medium were added to the upper chamber of a Boyden chamber insert coated with Matrigel (8-μm pore size filter) (BD Pharmingen, San Diego, CA). The lower chamber contained 10% FBS/DMEM/F12 (1:1) media. Cells were incubated for 40 h in a 37 °C, 5% CO_2_ incubator. Invasion of cells to the underside of the Matrigel-coated membrane was detected by fixing and staining the cells with 10% formalin and 0.5% Crystal Violet, respectively. Cells were counted under a microscope in three random fields/insert (magnification 100×).

### 4.13. PC3ML Glycolytic Rate Assay

50,000 PC3ML were seeded to a 24-well Seahorse XFe24 microplate and incubated overnight at 37 °C at 5%CO_2_; cells were 70–80% confluent at time of assay. The Agilent XFe24 Seahorse instrument was preheated to 37°C the day prior to assay. The Agilent hydrobooster plate (cat# 102342-100) was hydrated with 1mL calibrant (Cat3 103059-000) per well and incubated overnight at 37 °C 0% CO_2_ the day prior to assay. Glycolysis Rate Test Kit drugs Rotenone/Antimycin and 2-deoxy-D-glucose (Agilent, Santa Clara, CA, USA, cat# 103344-100) were allowed to come to room temperature prior to use. The day of the assay, 100 mL Agilent seahorse assay media (Cat# 102365-100) was prepared by the addition of 10 mM D-glucose, 2 nM glutamine, 1 mM pyruvate, and 5.0 mM HEPES. PC3MLs in Seahorse microplates were washed twice with seahorse assay media and cell adherence checked via brightfield microscopy. 500 mL assay media was added to each well, and the plate was incubated at 37 °C 0% CO_2_ for 45 min. Stock solutions of rotenone/antimycin and 2-deoxy-D-glucose were prepared according to manufacturer’s instructions and added to ports A and B of the hydrobooster plate, respectively. Final concentrations of drugs for the assay were 0.5 μM rotenone/antimycin and 50 mM 2-deoxy-D-glucose. The instrument was calibrated with XF calibrant solution prior to loading the Seahorse microplate containing cells. Analysis settings utilized were as follows: 3 min mixing, 2 min waiting, 3 min measuring for the duration of the assay. GlycoPer was normalized to total protein content in each well.

### 4.14. Statistical Analysis

Statistical analysis was performed using GraphPad Prism Software v7.03 (GraphPad Software, San Diego, CA, USA). Data is expressed as the mean ± standard deviation. Biological replicates for quantitative data comprised the average of three technical replicates. Normality and homogeneity of variance assumptions were evaluated prior to additional statistical analysis. Student’s *t*-test was performed for comparisons of two groups, and ANOVA was performed for comparisons of three or more groups. A Tukey’s HSD test was performed for post-hoc analysis of groups when utilizing ANOVA. The null hypothesis was rejected for *p*-value < 0.05.

## Figures and Tables

**Figure 1 ijms-24-01229-f001:**
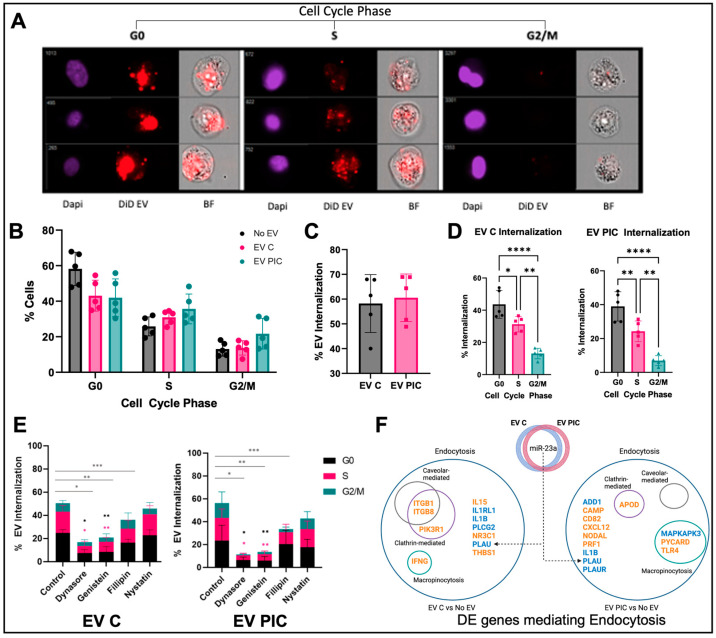
PC3ML EV internalization is cell cycle-dependent and independent of vesicle type. EV uptake and cell cycle phase was assessed using Amnis ImageStream. Representative images showing EV internalization (red), nuclei (blue), and phase contrast of cells in different cell cycle phases (**A**). The proportion of cells at each cell cycle phase did not change with EV C or EV PIC treatment relative to untreated control (**B**). Percent EV internalization was not different between EV C and EV PIC, *n* = 5 biological replicates, *p* = 0.7396 (**C**). Quantification of the proportion of EV internalization at each cell cycle phase in EV-treated cells revealed that the majority of internalization occurred at G0, followed by S phase, with the fewest vesicles being internalized at G2/M (EV C, *n* = 5, **** *p* < 0.001, * *p* = 0.0183, ** *p* = 0.0012; EV PIC, *n* = 5, **** *p* < 0.0001, ** *p* = 0.00027) (**D**). PC3MLs were incubated with endocytosis inhibitors dynasore, genistein, fillipin, or nystatin during vesicle treatment. In both EV C and EV PIC treated cells, EV internalization was significantly reduced by treatment with dynasore (EV C, *n* = 5, * *p* = 0.0001; EV PIC, *n* = 5, * *p* < 0.0001), genistein (EV C, *n* = 5, ** *p* = 0.0004; EV PIC, *n* = 5, ** *p* < 0.0001), and fillipin (EV C, *n* = 5, *** *p* = 0.021; EV PIC, *n* = 5, *** *p* = 0.0053). dynasore and genistein significantly reduced the proportion of EV internalized at G0 (EV C, *n* = 5, * *p* < 0.0001, ** *p* < 0.0001; EV PIC, *n* = 5, * *p* = 0.0017, ** *p* = 0.0012) and S phase (EV C, *n* = 5, * *p* = 0.0008, ** *p* = 0.012; EV PIC, *n* = 5, * *p* = 0.0059, ** *p* = 0.0153). Asterisks in black and pink directly above dynasore and genistein bars indicate significant differences in cell cycle phase proportion (**E**). Differential expression analysis of cells treated with EV C vs. control and cells treated with EV PIC vs. control showed deregulation of genes mediating multiple mechanisms of endocytosis (**F**). Genes in blue are downregulated, and genes in orange are upregulated.

**Figure 2 ijms-24-01229-f002:**
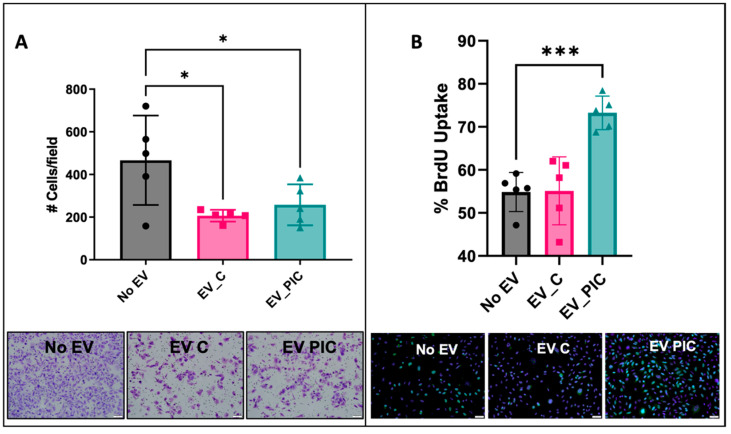
Effect of HAMVEC EV C and EV PIC on PC3ML invasion and proliferation. PC3MLs were incubated overnight with HAMVEC EV C or EV PIC, and invasion was assessed using Matrigel invasion chambers. PC3ML treated with HAMVEC EV C or EV PIC showed reduced invasion by 2.3- and 1.8-fold, respectively (EV C, *n* = 5, * *p* = 0.0147; EV PIC, *n* = 5, * *p* = 0.0405). Scale bar indicates 100 μm (**A**). Proliferation was assessed via fluorescent microscopy evaluation of BrdU incorporation. HAMVE EV PIC but not EV C increased proliferation by 1.3-fold (*n* = 5, *** *p* = 0.0005). Scale bar indicates 50 μm (**B**).

**Figure 3 ijms-24-01229-f003:**
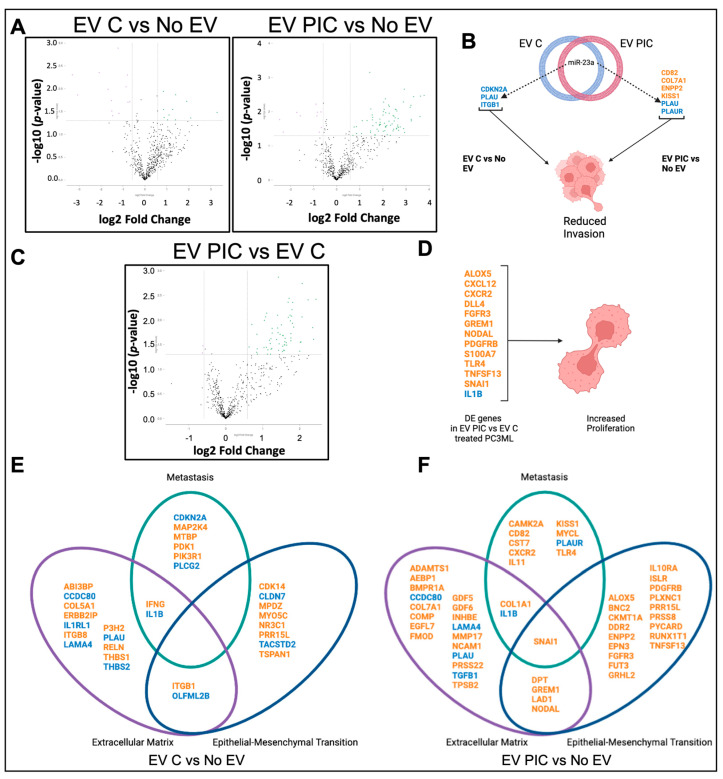
EV C and EV PIC deregulate genes involved in extracellular matrix remodeling, metastasis, and epithelial-mesenchymal transition. Volcano plots show DE genes with an x-axis indicating DE fold change with a threshold of +/− 0.58 log2 fold change, *p* < 0.05. Differential expression (DE) analysis revealed that 38 genes were deregulated with EV C treatment and 70 with EV PIC treatment relative to untreated control (**A**). IPA was used to identify DE genes involved in invasion in both EV C- and EV PIC-treated cells, and miRNA target filter analysis identified miR-23a as targeting *PLAU*, which is downregulated in both EV C- and EV PIC-treated cells as compared to untreated control (**B**). DE analysis of EV PIC-treated PC3MLs vs. EV C-treated revealed 55 DE genes (**C**). IPA analysis revealed 13 DE genes whose regulation status may have an impact on observed increased proliferation (**D**). Genes deregulated by EV C or EV PIC vs. untreated PC3MLs were clustered by NanoString Pan Cancer progression panel gene annotations. Of the 38 genes deregulated in EV C vs. untreated PC3MLs, 16 clustered to extracellular matrix remodeling, 10 clustered to epithelial-mesenchymal transition, and 10 clustered to metastasis (**E**). Of the 70 genes deregulated in EV PIC vs. untreated PC3ML, 25 clustered to extracellular matrix remodeling, 23 clustered to epithelial-mesenchymal transition, and 12 to metastasis (**F**). Genes in blue are downregulated, and genes in orange are upregulated.

**Figure 4 ijms-24-01229-f004:**
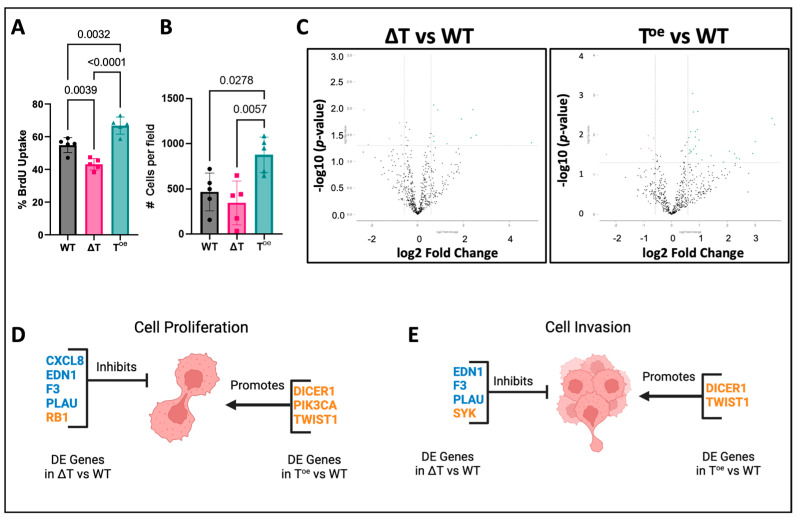
Endogenous *Twist1* influences the functional and molecular phenotype of PC3MLs. PC3MLs expressing either wild-type levels of *TWIST1* (WT), deficient (ΔT), or overexpressed (T^oe^) *TWIST1* were incubated with BrdU for 6hrs, and BrdU incorporation was scored via fluorescence microscopy. Proliferation was correlated with *TWIST1* expression; PC3MLs deficient in *TWIST1* had significantly lower proliferation than wild-type (*n* = 5), and PC3MLs overexpressing *TWIST1* had significantly increased proliferation (*n* = 5) (**A**). Invasion was assessed using Matrigel invasion chambers; overexpression of *TWIST1* significantly increased invasion in PC3MLs as compared to wild-type and cells deficient in *TWIST1* (*n* = 5) (**B**). Volcano plots show DE genes with an x-axis indicating DE fold change with a threshold of +/− 0.58 log2 fold change, *p* < 0.05. Differential expression (DE) analysis revealed 12 DE genes in ΔT vs. WT and 29 DE genes in T^oe^ vs. WT (**C**). IPA analysis revealed genes deregulated in ΔT or T^oe^ vs. WT PC3MLs whose regulation status may impact on observed changes in proliferation (**D**) or invasion (**E**). Genes in blue are downregulated, and genes in orange are upregulated.

**Figure 5 ijms-24-01229-f005:**
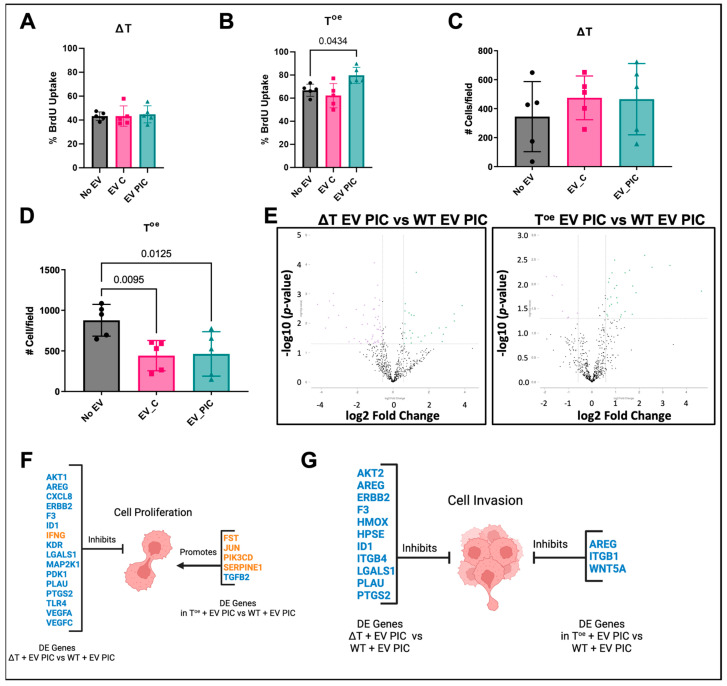
EV effect on PC3ML function is *TWIST1*-dependent. EV PIC had no effect on proliferation in ΔT PC3MLs but significantly increased proliferation in T^oe^ PC3MLs (*n* = 5) (**A**,**B**). Invasion was assessed using Matrigel invasion chambers; neither EV C nor EV PIC altered invasion in PC3MLs deficient in *TWIST1*, and both EV C and EV PIC reduced invasion in cells overexpressing *TWIST1* (EV C, *n* = 5, *p* = 0.0070; EV PIC, *n* = 5, *p* = 0.0092) (**C**,**D**). Volcano plots show DE genes with an x-axis indicating DE fold change with a threshold of +/− 0.58 log2 fold change, *p* < 0.05. The mRNA transcriptome of either ΔT or T^oe^ cells treated with EV PIC was compared to the transcriptome of WT cells treated with EV PIC. Differential expression (DE) analysis revealed 75 DE genes in ΔT + EV PIC vs. WT + EV PIC and 35 DE genes in T^oe^ + EV PIC vs. WT + EV PIC (**E**). IPA analysis revealed genes deregulated in ΔT + EV PIC vs. WT + EV PIC or T^oe^ + EV PIC vs. WT + EV PIC whose regulation status may impact on observed changes in proliferation (**F**) or invasion (**G**). Genes in blue are downregulated, and genes in orange are upregulated.

**Figure 6 ijms-24-01229-f006:**
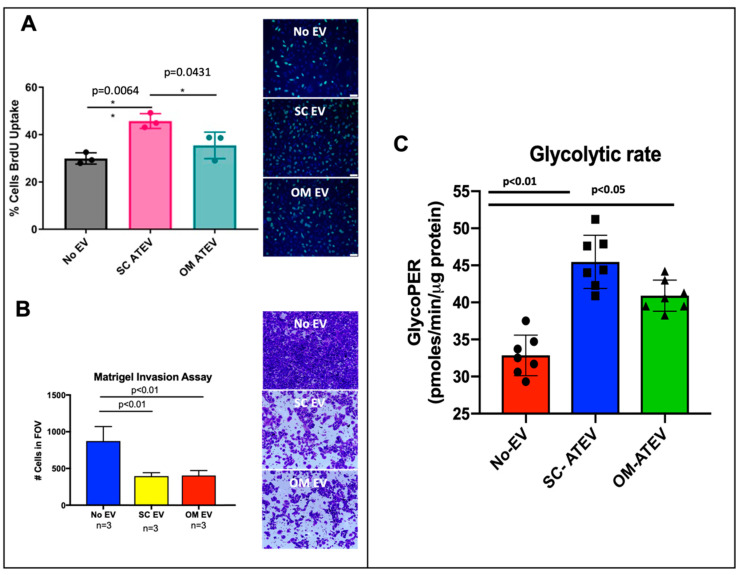
ATEV alters the functional signature of PC3ML. PC3MLs were incubated overnight with subcutaneous (SC) or omental (OM)ATEV, and proliferation was assessed via fluorescent microscopy evaluation of BrdU uptake. ATEV from subcutaneous and omental adipose tissue increased proliferation of PC3ML by 1.8-fold. Images taken with 200× magnification(**A**). Vesicles from both SC and OM adipose tissue depots reduced invasion of PC3ML by 2-fold. Images taken with 100× magnification (**B**). Glycolytic rate as measured using Agilent Seahorse analyzer was increased in cells treated with SC or OM ATEV (**C**). *, *p* < 0.05.

**Figure 7 ijms-24-01229-f007:**
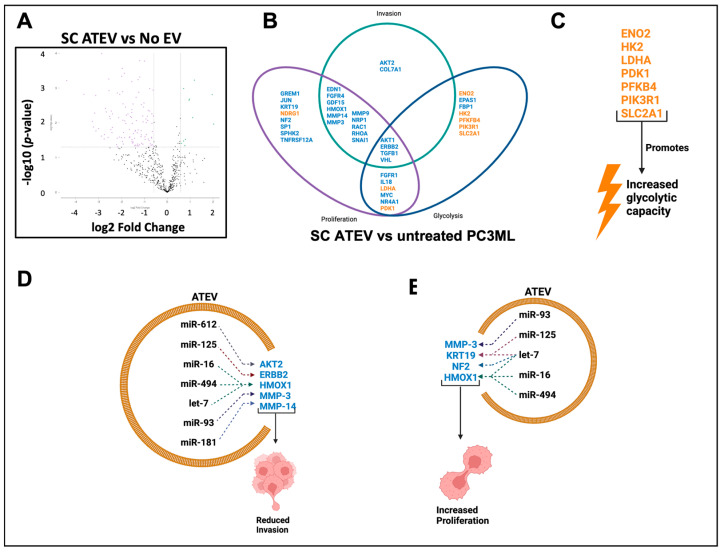
ATEV alters the molecular signature of PC3ML. Volcano plots show DE genes with an x-axis indicating DE fold change with a threshold +/− 0.58 log2 fold change, *p* < 0.05. SC ATEV changed expression of 108 genes (**A**). Of the 108 deregulated genes, 29 were associated with proliferation, 17 with invasion, and 17 with glycolysis (**B**). ATEV upregulated multiple genes that may impact on observed increase in glycolytic activity (**C**). miRNA contained within ATEV was found to target deregulated genes that may impact on observed reduction in invasion (**D**) and increased proliferation (**E**). Genes in blue are downregulated, and genes in orange are upregulated.

**Table 1 ijms-24-01229-t001:** miRNA mean values * detected in EV C and EV PIC.

miRNA Detected	Mean Expression Values (Counts)	
	EV C	EV PIC
hsa-miR-612	38.04	34.47
hsa-miR-126-3p	45.03	28.45
hsa-miR-302d-3p	58.50	55.58
hsa-miR-4454/miR-7975	35.76	26.80
hsa-miR-1253	28.06	37.53
hsa-miR-378e	33.23	42.35
hsa-miR-451a	38.70	32.95
hsa-miR-548ar-5p	27.68	27.08
hsa-miR-3144-3p	27.01	28.38
hsa-miR-23a-3p	26.33	21.84
hsa-miR-598-3p	21.66	20.19
hsa-miR-1246	28.78	25.64
hsa-miR-379-5p	23.34	29.69
hsa-miR-411-5p	21.29	24.74
hsa-miR-21-5p	24.77	29.00
hsa-let-7i-5p	31.80	38.31
hsa-miR-128-1-5p	20.00	20.87
hsa-miR-155-5p	20.00	22.83
hsa-miR-1285-5p	20.00	20.42
hsa-miR-579-3p	20.00	59.38

* Mean values represent normalized data from five biological replicates.

**Table 2 ijms-24-01229-t002:** Genes deregulated in EV C-treated versus untreated PC3MLs.

Gene	log2 Fold Change	*p*-Value	Gene	log2 Fold Change	*p*-Value
*NOX5*	3.07	0.004639	*CDK14*	0.66	0.004271
*IFNG*	2.81	0.03712	*ITGB1*	0.63	0.027101
*RELN*	2.08	0.033885	*MAP2K4*	0.62	0.006884
*PRR15L*	2.02	0.048645	*RBPJ*	0.61	0.034749
*ABI3BP*	1.77	0.002004	*ERBB2IP*	0.58	0.026196
*ITGB8*	1.75	0.046099	*CDKN2A*	−0.86	0.007913
*TYMP*	1.75	0.031495	*THBS2*	−0.87	0.013662
*TSPAN1*	1.64	0.028679	*JUN*	−0.95	0.006642
*THBS1*	1.24	0.007253	*PLAU*	−1.01	0.00659
*IL15*	0.98	0.041996	*CLDN7*	−1.04	0.044866
*NR3C1*	0.96	0.014767	*IL1RL1*	−1.30	0.020427
*MYO5C*	0.94	0.002495	*IL1B*	−1.36	0.04731
*COL5A1*	0.88	0.027718	*TACSTD2*	−1.55	0.000126
*PIK3R1*	0.74	0.017385	*OLFML2B*	−1.67	0.010756
*PDK1*	0.73	0.046052	*ROBO4*	−1.71	0.018305
*MPDZ*	0.68	0.038911	*PLCG2*	−1.72	0.012757
*MTBP*	0.68	0.008062	*LAMA4*	−1.90	0.000202
*DICER1*	0.68	0.035538	*CCDC80*	−2.16	0.001262
*P3H2*	0.66	0.047682	*TIE1*	−3.50	0.001308

**Table 3 ijms-24-01229-t003:** Genes deregulated in EV PIC-treated versus untreated PC3MLs.

Gene	log2 Fold Change	*p* Value	Gene	log2 Fold Change	*p* Value
*PRR15L*	3.79	0.0034	*COL7A1*	1.90	0.0237
*AEBP1*	3.63	0.0035	*FGFR3*	1.88	0.0204
*GDF6*	3.33	0.0036	*TNFSF13*	1.87	0.0166
*DDR2*	3.23	0.0169	*ALOX5*	1.86	0.0236
*DPT*	3.22	0.0065	*FMOD*	1.86	0.0248
*S100A7*	3.18	0.0074	*NCAM1*	1.85	0.0250
*COMP*	2.93	0.0022	*PRF1*	1.81	0.0371
*TPSB2*	2.90	0.0314	*ISLR*	1.75	0.0327
*CAMK2A*	2.90	0.0052	*PRSS22*	1.71	0.0211
*CAMP*	2.65	0.0247	*ADAMTS12*	1.66	0.0487
*INHBE*	2.65	0.0277	*CXCR3*	1.57	0.0258
*CXCL12*	2.61	0.0096	*KISS1*	1.56	0.0146
*CXCR2*	2.60	0.0320	*EPHB3*	1.55	0.0174
*PRSS8*	2.57	0.0086	*MMP17*	1.54	0.0125
*DLL4*	2.49	0.0188	*EPN3*	1.48	0.0084
*BAI1*	2.48	0.0136	*APOD*	1.47	0.0367
*TLR4*	2.47	0.0326	*ENPP2*	1.47	0.0411
*PPP1R16B*	2.44	0.0130	*COL1A1*	1.44	0.0007
*PLXNC1*	2.40	0.0228	*BAI3*	1.32	0.0182
*CHI3L1*	2.33	0.0249	*CD82*	1.11	0.0185
*EGFL7*	2.28	0.0070	*MYCL*	1.08	0.0438
*RUNX1T1*	2.27	0.0112	*CKMT1A*	0.86	0.0436
*NODAL*	2.27	0.0190	*BNC2*	0.85	0.0276
*IL10RA*	2.25	0.0064	*SNAI1*	0.75	0.0282
*PYCARD*	2.17	0.0255	*BMPR1A*	0.70	0.0495
*GRHL2*	2.13	0.0278	*ADD1*	−0.67	0.0293
*GDF5*	2.13	0.0108	*PLAUR*	−0.68	0.0102
*TBX4*	2.12	0.0268	*MAPKAPK3*	−0.75	0.0397
*LRG1*	2.12	0.0285	*PLAU*	−0.84	0.0104
*IL11*	2.11	0.0042	*JUN*	−0.85	0.0134
*CST7*	2.07	0.0219	*IL1B*	−0.89	0.0060
*LAD1*	2.06	0.0086	*LAMA4*	−1.58	0.0133
*FUT3*	2.03	0.0053	*CCDC80*	−1.72	0.0106
*GREM1*	2.03	0.0100	*TGFBI*	−2.32	0.0397
*PDGFRB*	2.01	0.0126	*TIE1*	−2.47	0.0172

**Table 4 ijms-24-01229-t004:** Genes deregulated in EV PIC- versus EV C-treated PC3MLs.

Gene	log2 Fold Change	*p*-Value	Gene	log2 Fold Change	*p*-Value
*CHI3L1*	2.37	0.024684	*ADAMTS12*	1.43	0.041283
*DPT*	2.35	0.01427	*CCL21*	1.42	0.001353
*GDF6*	2.22	0.003901	*DLL4*	1.41	0.01806
*PPP1R16B*	2.07	0.001819	*GRHL2*	1.40	0.014536
*S100A7*	2.06	0.008665	*IL10RA*	1.40	0.008006
*LRG1*	2.01	0.028425	*RUNX1T1*	1.25	0.018417
*PRR15L*	1.84	0.013931	*TBX4*	1.25	0.014792
*FMOD*	1.81	0.004403	*VIT*	1.24	0.037636
*NCAM1*	1.80	0.011674	*PRSS22*	1.22	0.0392
*EGFL7*	1.79	0.006192	*CXCR3*	1.22	0.024046
*FUT3*	1.78	0.008341	*CLEC3B*	1.20	0.029903
*AEBP1*	1.76	0.02649	*PLXNC1*	1.18	0.025741
*COMP*	1.75	0.010389	*ENPP2*	1.14	0.040427
*PDGFRB*	1.74	0.007964	*AQP1*	1.13	0.049259
*ISLR*	1.71	0.003647	*TMEM30B*	1.11	0.007273
*KISS1*	1.69	0.008803	*MMP3*	1.10	0.035844
*IL11*	1.65	0.032415	*EPHB3*	1.08	0.008622
*DDR2*	1.64	0.028908	*BAI3*	1.04	0.010188
*MMRN2*	1.60	0.02365	*EPN3*	1.03	0.036403
*GIMAP4*	1.60	0.027679	*MMP9*	0.85	0.049227
*GDF5*	1.58	0.032004	*ECM2*	0.85	0.038402
*BAI1*	1.57	0.041078	*FBLN5*	0.84	0.019496
*CAMK2A*	1.55	0.020639	*ECM1*	0.84	0.045216
*ERBB3*	1.54	0.034159	*COL1A1*	0.70	0.01234
*GREM1*	1.54	0.020635	*PLXND1*	0.63	0.035244
*ALOX5*	1.53	0.043804	*TPM2*	−0.62	0.033518
*KLK3*	1.50	0.012539	*PPL*	−0.63	0.046751
*LAD1*	1.47	0.019572			

**Table 5 ijms-24-01229-t005:** Differentially expressed genes in ΔT versus WT.

ΔT No EV vs. WT No EV		
Gene Name	log2 Fold Change	*p*-Value
*SYK*	4.97	0.040658
*IL11*	1.98	0.013309
*ITGB2*	1.31	0.041964
*MPDZ*	0.87	0.01136
*SOX9*	0.69	0.046489
*RB1*	0.63	0.025716
*PLAU*	−0.71	0.048369
*F3*	−0.73	0.031497
*SHB*	−0.8	0.0407
*EDN1*	−1.19	0.044181
*ANGPTL4*	−2.32	0.011077
*CXCL8*	−2.58	0.023576

**Table 6 ijms-24-01229-t006:** Differentially expressed in T^oe^ versus WT.

T^oe^ No EV vs. WT No EV		
	log2 Fold Change	*p*-Value
*TWIST1*	3.6	0.00384133
*CHRDL1*	3	0.01478225
*PRR15L*	2.25	0.02989221
*COMP*	2.07	0.02586877
*BAI1*	1.86	0.048146
*AGR2*	1.42	0.03341637
*DST*	1.09	0.04268855
*CD2AP*	0.98	0.01342084
*PIK3R1*	0.94	0.00257603
*MPDZ*	0.93	0.00735395
*ITGB3*	0.89	0.03016183
*SMC3*	0.86	0.01915108
*DICER1*	0.85	0.00827682
*POPDC3*	0.85	0.01136584
*VPS13A*	0.79	0.02277646
*DPYSL3*	0.78	0.00266879
*BAG2*	0.76	0.00090347
*SMAD1*	0.7	0.02558088
*PIK3CA*	0.69	0.02713395
*MTBP*	0.68	0.00837542
*SMURF2*	0.65	0.02394016
*PLS1*	0.63	0.02915665
*FLI1*	0.62	0.01174225
*SORD*	−0.65	0.01201018
*TMC6*	−0.66	0.04140682
*TNFRSF1A*	−0.81	0.02165432
*ITGA3*	−0.83	0.01050531
*HSPG2*	−1.1	0.022335
*JAM3*	−2.34	0.0312228

**Table 7 ijms-24-01229-t007:** Differentially expressed genes in ΔT + EV PIC versus WT + EV PIC.

Gene	log2 Fold Change	*p*-Value	Gene	log2 Fold Change	*p*-Value
*HAPLN1*	−4.21	0.00229681	*AKT1*	−0.82	0.00585506
*CEACAM6*	−3.77	0.01457229	*F3*	−0.81	0.01620524
*PTGS2*	−3.58	0.00651015	*ADAM15*	−0.80	0.04358126
*CXCL8*	−3.54	0.00175103	*ENO1*	−0.77	0.00246709
*ITM2A*	−3.32	0.00096453	*EMP3*	−0.74	0.00990741
*KDR*	−3.32	0.00248596	*PPP2R1A*	−0.74	0.00162897
*JAM3*	−3.03	0.00535929	*HDAC5*	−0.72	0.01330461
*RELN*	−2.45	0.04281032	*NME4*	−0.71	0.0168286
*ELF3*	−2.16	0.03343441	*VEGFC*	−0.64	0.02863055
*ID1*	−2.05	0.00337365	*MAP2K1*	−0.62	0.01808655
*ANGPTL4*	−1.87	0.04213076	*HDHD3*	−0.59	0.03844262
*NDRG1*	−1.79	0.049126	*ALDOA*	−0.59	0.02255396
*BMPR1B*	−1.72	0.01115635	*ROCK1*	0.62	0.049729
*RRAS*	−1.65	0.00521842	*RB1*	0.65	0.02119552
*CYB561*	−1.58	0.00241057	*MTBP*	0.66	0.0104928
*ROBO4*	−1.52	0.03596858	*BAG2*	0.66	0.00380487
*AREG*	−1.43	0.01053543	*KRAS*	0.68	0.00216705
*EGLN3*	−1.43	0.04298111	*SOX9*	0.74	0.03225364
*PLCG2*	−1.40	0.0407531	*PIK3R1*	0.89	0.00445334
*ITGB4*	−1.31	0.049818	*COL5A1*	0.94	0.01783749
*BAD*	−1.18	0.03426514	*SACS*	1.00	0.03168268
*ERBB2*	−1.14	0.02879555	*VEGFA*	1.02	0.04694106
*FBN1*	−1.12	0.02488784	*PDK1*	1.02	0.00523768
*PLAU*	−1.10	0.00334384	*BNC2*	1.08	0.02162339
*HMOX1*	−1.10	0.04775754	*JAG1*	1.17	0.00548071
*EGLN2*	−1.05	8.78 × 10^−5^	*MPDZ*	1.29	0.00018507
*HPSE*	−1.03	0.00324885	*ITGB2*	1.40	0.02970008
*CAMK2D*	−1.03	0.01194406	*SCNN1A*	1.56	0.04077433
*LAMA4*	−1.02	0.04320141	*IL11*	1.77	0.02678524
*LGALS1*	−1.00	0.00136589	*COMP*	2.14	0.02144892
*PDPN*	−0.99	0.04515718	*PRR15L*	2.57	0.01449344
*TNFRSF1A*	−0.97	0.00626676	*IFNG*	2.74	0.0417943
*PPFIBP2*	−0.89	0.04276648	*ADAMTS12*	2.80	0.02355363
*AKT2*	−0.89	0.01584298	*AEBP1*	2.95	0.01395774
*MAPKAPK3*	−0.86	0.00751957	*FREM2*	3.41	0.00480102
*SERINC5*	−0.85	0.00050478	*TLR4*	3.42	0.00836757
*TNFSF12*	−0.84	0.03804925	*NCAM1*	3.85	0.00251067

**Table 8 ijms-24-01229-t008:** Differentially expressed genes in T^oe^ + EV PIC versus WT + EV PIC.

Gene	log2 Fold Change	*p*-Value	Gene	log2 Fold Change	*p*-Value
*TGFBI*	4.64	0.01391693	*JUN*	0.82	0.0258561
*TIE1*	3.3	0.0041821	*AKAP12*	0.8	0.0183849
*ARHGBIB*	2.55	0.00453406	*POPDC3*	0.75	0.0279922
*COL6A3*	2.23	0.00259815	*P3H1*	0.73	0.0076634
*DSC2*	1.87	0.01504002	*TGFBR2*	0.7	0.0091271
*CCDC80*	1.73	0.01096193	*SMAD1*	0.67	0.0202257
*LY96*	1.7	0.04121135	*SNAI2*	0.67	0.0447615
*AKT3*	1.59	0.00522403	*DPYSL3*	0.63	0.0413461
*LAMC2*	1.4	0.03640135	*ITGB1*	−0.61	0.0391456
*SERPINE1*	1.4	0.00589652	*TIMP1*	−0.99	0.047494
*SPOCK3*	1.24	0.04244838	*WNT5A*	−1.02	0.0488749
*IL1RL1*	1.21	0.0468779	*COL5A2*	−1.24	0.0358520
*PIK3CD*	1.07	0.02407889	*FST*	−1.26	0.0104353
*TJP2*	1.05	0.01906991	*CTSH*	−1.5	0.0070984
*TNC*	1.05	0.01068077	*AREG*	−1.6	0.0068397
*ITGA5*	0.96	0.00325054	*TGFB2*	−1.68	0.0171156
*F11R*	0.95	0.01216205	*ID2*	−1.93	0.0070852
*COL4A1*	0.85	0.00590717			

**Table 9 ijms-24-01229-t009:** miRNA detected in SC and OM ATEV.

Probe Name	SC Mean * Counts	OM Mean Counts	Probe Name	SC Mean Counts	OM Mean Counts
hsa-miR-612	27.13	22.26	hsa-miR-122-5p	72.38	72.46
hsa-miR-195-5p	22.20	24.46	hsa-miR-99a-5p	68.71	72.88
hsa-miR-30e-5p	22.18	24.59	hsa-let-7c-5p	61.60	76.96
hsa-miR-191-5p	23.63	25.99	hsa-miR-16-5p	238.04	100.80
hsa-miR-93-5p	26.05	27.26	hsa-miR-29b-3p	132.43	102.48
hsa-miR-29a-3p	29.11	27.48	hsa-miR-199b-5p	177.47	104.99
hsa-miR-374a-5p	32.29	28.97	hsa-miR-144-3p	730.53	120.83
hsa-miR-365a/b	25.93	29.06	hsa-let-7b-5p	162.87	132.40
hsa-let-7d-5p	22.62	32.37	hsa-let-7a-5p	123.25	163.11
hsa-miR-376a-3p	28.78	33.25	hsa-miR-199a/b	328.73	163.79
hsa-miR-497-5p	33.85	34.06	hsa-miR-23a-3p	401.38	215.57
hsa-miR-15b-5p	83.20	35.55	hsa-miR-125b-5p	306.70	269.68
hsa-miR-25-3p	67.31	35.63	hsa-miR-445/miR-7975	1193.29	744.56
hsa-miR-15a-5p	61.66	35.91	hsa-miR-451a	5206.46	968.58
hsa-miR-26b-5p	34.30	36.42	hsa-miR-363-3p	20.35	*
hsa-miR-150-5p	36.62	38.45	hsa-miR-129-5p	26.70	*
hsa-miR-320e	33.18	39.11	hsa-miR-30c-5p	36.47	*
hsa-miR-145-5p	35.25	39.17	hsa-miR-223-3p	22.85	*
hsa-miR-630	47.80	41.89	hsa-miR-378g	*	22.71
hsa-miR-21-5p	46.09	42.68	hsa-miR-214-3p	*	23.13
hsa-miR-142-3p	70.88	43.94	hsa-miR-498	*	23.68
hsa-miR-4286	35.72	44.62	hsa-miR-574-3p	*	25.45
hsa-miR-148a-3p	34.88	46.31	hsa-miR-181a-5p	*	25.78
hsa-miR-100-5p	39.07	46.85	hsa-miR-342-3p	*	25.88
hsa-miR-199a-5p	51.93	48.63	hsa-miR-411-5p	*	27.27
hsa-miR-22-3p	45.90	48.73	hsa-miR-23b-3p	*	31.57
hsa-let-7i-5p	42.12	50.77	hsa-miR-128-1-5p	*	32.41
hsa-miR-130a-3p	26.32	53.22	hsa-miR-10a-5p	*	32.76
hsa-let-7g-5p	53.85	55.04	hsa-miR-494-3p	*	64.51
hsa-miR-126-3p	45.07	60.14			

* Mean values represent normalized data from three biological replicates.

**Table 10 ijms-24-01229-t010:** miRNA detected in both ATEV and HAMVEC EV C and EV PIC.

	Mean * Counts			
Probe Name	SC ATEV	OM ATEV	EV C	EV PIC
hsa-let-7i-5p	42.12	50.77	31.80	38.31
hsa-miR-126-3p	45.07	60.14	45.03	28.45
hsa-miR-128-1-5p	20.34	32.41	20.00	20.87
hsa-miR-21-5p	46.09	42.68	24.77	29.00
hsa-miR-23a-3p	401.38	215.57	26.33	21.84
hsa-miR-411-5p	20.49	27.27	21.29	24.74
hsa-miR-4454/miR-7975	1193.29	744.56	35.76	26.80
hsa-miR-451a	5206.46	968.58	38.70	32.95
hsa-miR-612	27.13	22.26	38.04	34.47

* Mean values represent normalized data from three biological replicates.

**Table 11 ijms-24-01229-t011:** Genes deregulated in ATEV-treated vs. untreated PC3MLs.

Gene	log2 Fold Change	*p* Value	Gene	log2 Fold Change	*p* Value
*NDRG1*	2.03	0.0109	*TGFB1*	−1.31	0.0195
*PFKFB4*	1.97	0.0015	*MMP14*	−1.32	0.0451
*HK2*	1.19	0.0006	*LAMA4*	−1.35	0.0087
*SLC2A1*	1.11	0.0071	*LHFP*	−1.36	0.0019
*ENO2*	0.97	0.0021	*RNH1*	−1.39	0.0166
*PIK3R1*	0.95	0.0022	*EDN1*	−1.40	0.0173
*PDK1*	0.80	0.0301	*OLFML2B*	−1.43	0.0280
*MTDH*	0.77	0.0455	*SERPINH1*	−1.48	0.0206
*LDHA*	0.72	0.0010	*CCBE1*	−1.49	0.0359
*LIFR*	0.71	0.0335	*LAMA5*	−1.49	0.0029
*DENNDA*	0.61	0.0392	*ESRP1*	−1.50	0.0358
*AKT1*	−0.59	0.0475	*TNXB*	−1.53	0.0105
*RAC1*	−0.60	0.0064	*KRT19*	−1.54	0.0303
*RHOA*	−0.61	0.0011	*EPHB4*	−1.54	0.0041
*B3GNT3*	−0.65	0.0178	*TCF3*	−1.55	0.0344
*TMC6*	−0.65	0.0432	*BAD*	−1.55	0.0050
*TFDP1*	−0.67	0.0105	*NRP2*	−1.55	0.0007
*VHL*	−0.67	0.0055	*EVPL*	−1.56	0.0070
*SDC4*	−0.68	0.0262	*PIK3R2*	−1.61	0.0066
*UBA52*	−0.68	0.0273	*KRT7*	−1.63	0.0253
*TNFRSFA*	−0.70	0.0442	*ECM1*	−1.65	0.0155
*FGFR1*	−0.74	0.0402	*HSPB1*	−1.66	0.0269
*NRP1*	−0.77	0.0010	*CKMT1A*	−1.72	0.0056
*EIF4E2*	−0.77	0.0046	*COL1A1*	−1.78	0.0060
*AKT2*	−0.79	0.0327	*MMP3*	−1.81	0.0105
*IL18*	−0.80	0.0476	*CRISPLD2*	−1.82	0.0035
*SP1*	−0.83	0.0294	*LTBP4*	−1.85	0.0088
*HDAC5*	−0.83	0.0044	*COL18A1*	−1.87	0.0004
*MYC*	−0.85	0.0085	*FBP1*	−1.91	0.0190
*NF2*	−0.87	0.0065	*MMP9*	−1.93	0.0084
*RAC2*	−0.92	0.0218	*CRIP2*	−1.93	0.0028
*VAV2*	−0.92	0.0099	*LLGL2*	−1.95	0.0013
*JUN*	−0.93	0.0078	*HMOX1*	−1.96	0.0003
*SH2D3A*	−0.97	0.0300	*SNAI1*	−1.99	0.0005
*ETV4*	−0.97	0.0253	*VASH1*	−2.07	0.0015
*GPX1*	−0.99	0.0002	*GDF15*	−2.12	0.0072
*ANXA2P2*	−1.02	0.0027	*AGRN*	−2.18	0.0014
*SPHK2*	−1.04	0.0035	*GRHL2*	−2.19	0.0278
*RPS6KB2*	−1.05	0.0094	*CLEC3B*	−2.45	0.0046
*ERBB2*	−1.06	0.0415	*TNFRSF12A*	−2.46	0.0157
*ADAM15*	−1.07	0.0064	*NTRK1*	−2.50	0.0076
*PLXND1*	−1.12	0.0390	*AMH*	−2.52	0.0065
*WARS*	−1.13	0.0030	*TBXA2R*	−2.54	0.0012
*PKN1*	−1.14	0.0222	*RAMP1*	−2.55	0.0005
*PNPLA6*	−1.14	0.0168	*GREM1*	−2.58	0.0264
*CLDN3*	−1.15	0.0250	*COL7A1*	−2.66	0.0049
*AAMP*	−1.20	0.0005	*ECM2*	−2.70	0.0039
*ITGA3*	−1.20	0.0002	*ADAMTS12*	−2.74	0.0219
*FGFR4*	−1.22	0.0432	*NR4A1*	−2.86	0.0001
*SHB*	−1.25	0.0013	*CAMK2A*	−3.12	0.0172
*PKNOX1*	−1.27	0.0465	*CHP1*	−3.13	0.0016
*SLC37A1*	−1.27	0.0161	*TBX4*	−3.17	0.0052
*ZCCHC24*	−1.27	0.0006	*RUNX1T1*	−3.64	0.0015
*EPAS1*	−1.29	0.0409	*LAD1*	−4.39	0.0006

## Data Availability

All NanoString miRNA and mRNA data is available in the NCBI Gene Expression Omnibus data repository, accession GSE208209.

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
