# Peer review of "Adipose Tissue-Derived Extracellular Vesicles Contribute to Phenotypic Plasticity of Prostate Cancer Cells"

_ijms, 2023, doi:10.3390/ijms24021229_

Round 1

Reviewer 1 Report

In the study ‘Adipose Tissue-derived…..cancer cells’ by Mathiesen et al the authors investigate the impact of extracellular vesicles and endothelial cells derived from different sources of human adipose tissue on prostate cancer cells The study is well designed and in-depth relevant mechanistic investigations are carried out to provide scientific evidence.

However, there are some minor comments that the authors need to address:

1)      Given that the focus of the Ms. is on PC3Ml cell line, please provide more details about its tumor/metastatic origin and characteristics.

2)      As per the methods sections-the source of adipose tissue was both obese type-2 diabetic and non-diabetic patients, however, the study results do not define what is the influence of these variants? Please deliberate on this aspect.

3)      The subcutaneous and omental adipose tissue have different characteristics-with omental being more related to visceral obesity and directly contributing to systemic obesity while subcutaneous adipose depots may not directly influence systemic parameters or be representative of adipose tissue in the prostate vicinity, as such the authors may want to deliberate more on this in the discussion section.

4)      Fig. 2A, the representative images for invasion do not correlate with the data shown-please show images more representative of the data.

5)      Overall, in different figures the labeling on the volcano plots is not legible, please provide clear pictures.

Author Response

Response to Reviewer #1

The authors thank the reviewer for their careful reading of the manuscript and valuable feedback and comments.

  1. Given that the focus of the Ms. is on PC3Ml cell line, please provide more details about its tumor/metastatic origin and characteristics.

Reviewer 1 is correct, and we agree that a discussion on the origin and characteristics of the highly metastatic PC3-ML prostate cancer cell line. We have added a description of the cells in the introduction (page 1, lines 39-51)

  1. As per the methods sections-the source of adipose tissue was both obese type-2 diabetic and non-diabetic patients, however, the study results do not define what is the influence of these variants? Please deliberate on this aspect.

Thank you for the helpful feedback. The goal of these experiments was to compare the influence of fat depots specifically and as such, we did not evaluate the differential effects of the diabetic state of the patient. Samples from a limited number of patients were included in these experiments and therefore stratification for T2DM contribution could not be done due to lack of statistical power. We concede that this is a limitation of our study and have added a short statement addressing this limitation and potential future directions in the discussion section (page 23, lines 632-635)

  1. The subcutaneous and omental adipose tissue have different characteristics-with omental being more related to visceral obesity and directly contributing to systemic obesity while subcutaneous adipose depots may not directly influence systemic parameters or be representative of adipose tissue in the prostate vicinity, as such the authors may want to deliberate more on this in the discussion section.

Reviewer 1 makes an excellent point and we have added a section within the discussion to address the roles of omental and subcutaneous adipose tissue in relation to prostate cancer progression (page 23, lines 615-626)

  1. 2A, the representative images for invasion do not correlate with the data shown-please show images more representative of the data.

Thank you for the suggestion. Images for figure 2A updated to be more representative of the data shown.

  1. Overall, in different figures the labeling on the volcano plots is not legible, please provide clear pictures.

We agree that the labeling on the volcano plots is not legible. Axis parameters were added in larger font to each volcano plot and p-value and threshold parameter descriptions were added to each figure legend.

Reviewer 2 Report

Dear authors of “Adipose Tissue-Derived Extracellular Vesicles Contribute to Phenotypic Plasticity of Prostate Cancer Cells”,

Thanks for your contribution on this field. This is an interesting article, well organized and the research is well conducted and constructed. The manuscript is well written and organized. Nevertheless, there are some points which need to be solved before acceptance for publication.

Major points.

1-      The authors claim that there is a link between obesity and prostate cancer. This point is still open to debate. The reference points by the authors about this matter is related to periodontal disease and cancer. Even if a link between obesity and BPH is clearly define, it is not the case with prostate cancer. The authors should extend the bibliographic data in the introduction part.

2-      The authors demonstrated that PIC EVs have a dual effect on prostate cancer cells. In on hand they can upregulated cell proliferation and in the other, inhibit cell invasion. It will be of interest if the authors discuss about this dual and opposite effect. Indeed, very often both occur at the same time., especially when cell undergo on EMT process. In the same way, some genes are regulated as the drawn conclusions but some time it is not the case. For example, MMP-3 is well known to participate to cell tumorigenesis and invasion. Here this enzyme seems to be up-regulated which contradictory with the inhibition of cell invasion. Discussions about similar points will be of interest.

3-      One limit of this study is the use of PC3-LM cells. This cell line has already undergone to metastasis process. If PIC EVs are involved in the metastasis on prostate cancer cells, it will be more relevant to do such analysis with regular PC3 or DU145 cells and also a less aggressive one such as 22RV21.

 All the best,

Author Response

Response to Reviewer #2

We thank the reviewer for the valuable critiques and suggestions.

  1. The authors claim that there is a link between obesity and prostate cancer. This point is still open to debate. The reference points by the authors about this matter is related to periodontal disease and cancer. Even if a link between obesity and BPH is clearly define, it is not the case with prostate cancer. The authors should extend the bibliographic data in the introduction part.

Reviewer 1 is correct in that the link between obesity and prostate cancer is complex and not yet fully understood. We are suggesting that obesity-associated inflammation may play a role in prostate cancer progression. There is evidence to suggest that extracellular vesicles released from the proinflammatory obese adipose environment may promote prostate cancer progression and I have added a section in the introduction to more clearly describe this relationship. We have added clarification to this point (page 2, lines 31-32, 36-37 and page 3 lines 72-89).

  1. The authors demonstrated that PIC EVs have a dual effect on prostate cancer cells. In on hand they can upregulated cell proliferation and in the other, inhibit cell invasion. It will be of interest if the authors discuss about this dual and opposite effect. Indeed, very often both occur at the same time., especially when cell undergo on EMT process. In the same way, some genes are regulated as the drawn conclusions but some time it is not the case. For example, MMP-3 is well known to participate to cell tumorigenesis and invasion. Here this enzyme seems to be up-regulated which contradictory with the inhibition of cell invasion. Discussions about similar points will be of interest.

The dual effect is of EV PIC upregulating proliferation and inhibiting invasion is consistent with the phenotypic shift that occur during mesenchymal-to-epithelial transition; the cells shift from a less proliferative, more invasive phenotype to a more proliferative less invasive phenotype. This enables the colonization of secondary tumor sites. While both MMP3 and MMP9 are upregulated in EV PIC treated cells, upregulation alone does not always promote invasion. CD82 is also upregulated in EV PIC treated PC3ML and has been shown to suppress invasion by inactivating MMP9. Also of note is the downregulation of plasminogen genes PLAU and PLAUR which are important pro-MMP activators. We have added these considerations to the discussion (page 7 lines 217-221, page 21, lines 505-13, references 60-63).

  1. One limit of this study is the use of PC3-LM cells. This cell line has already undergone to metastasis process. If PIC EVs are involved in the metastasis on prostate cancer cells, it will be more relevant to do such analysis with regular PC3 or DU145 cells and also a less aggressive one such as 22RV21.

The effect of EV on the highly metastatic PC3ML which models more advanced and aggressive disease is less studied than those of less malignant phenotypes. The effect of obese adipose-associated EV have been previously studied in PC3, DU145, and 22RV1 cells. We sought to address this gap in knowledge by exploring the effects of obese adipose-derived EV on a more advanced and aggressive phenotype. We have added a short paragraph addressing this limitation to the discussion (Page 23, lines 627-631, references 26 and 110-114).

Reviewer 3 Report

The authors tried to investigate the role of adipose tissue-derived extracellular vesicles on the phenotypic plasticity of Prostate Cancer Cells. Although this study provides important results, the authors need to address several points to improve this article. 

#1 Introduction needs to be shortened and simplified.  Please demonstrate 1. what is known, 2, what is unknown, and 3. what the authors did to improve it.

#2 The authors need to show this phenomenon in patients with prostate cancer (ex. patients with obesity vs. skinny). Cellular experiments alone are a theoretical exercise. 

#3 Why the authors focused on Twist1 alone? There are many potential candidates. It is hard to believe that Twist1 is the only one involved in the phenotypic plasticity of prostate cancer. 

#4 Discrepancy between proliferation and invasion seems strange. In Fig.4, TWIST1 promotes both proliferation and invasion. Please clarify it. 

Author Response

Response to Reviewer #3

We thank you for your thorough review and constructive comments.

  1. Introduction needs to be shortened and simplified.  Please demonstrate 1. what is known, 2, what is unknown, and 3. what the authors did to improve it.

The introduction was modified for clarity; specifically the paragraph describing the link between obesity and prostate cancer has was simplified and shortened to more clearly describe what is known about the relationship between inflammation, obesity and prostate cancer.

  1. The authors need to show this phenomenon in patients with prostate cancer (ex. Patients with obesity vs. skinny). Cellular experiments alone are a theoretical exercise.

Translational approaches are the ultimate goal for these types of studies; however this study provides a proof-of-concept that lays the foundation for future translational work.

  1. Why the authors focused on Twist1 alone? There are many potential candidates. It is hard to believe that Twist1 is the only one involved in the phenotypic plasticity of prostate cancer.

Twist1 is certainly not the only factor, however it is a key driver and has a cascading effect that impacts on numerous pro-oncogenic functions. We chose to focus specifically on Twist1 because it is upregulated in prostate cancer and its expression is correlated with high Gleason scores and because of its key role in EMT and numerous downstream targets. IN addition, PC3 cells were the primary biologic material used for cloning of the human Twist-1 gene, due to the robust expression of the latter. I have added a brief description of complementary transcription factors that act in concert with Twist1 and our reasoning for focusing on Twist1 (page 3, lines 100-108, references 42-44)

  1. Discrepancy between proliferation and invasion seems strange. In Fig. 4, Twist1 promotes both proliferation and invasion. Please clarify it.

While it may seem counterintuitive for Twist1 to promote both proliferation and invasion in the Twist1 overexpressing cells, it is not uncommon to see this phenotypic flexibility in vitro. Highly metastatic cancer cells may shift from a proliferative to invasive phenotype as they adapt to encountered stimuli. It is also important to note that the T cells are only a single allele knockdown which may have produced a hybrid epithelial-mesenchymal phenotype. We have added brief explanations to address this point (page 9 lines 260-262, page 11-12, lines 314-317 and page 23, lines 596-598)

Round 2

Reviewer 2 Report

Dear authors,  

Thanks for considering my raised from the first submission. 

Now, the manuscript can be accepted for publication. 

Congratulations, 

All the best... 

Reviewer 3 Report

The reviewer agree with the revisions.